# Sensing leg movement enhances wearable monitoring of energy expenditure

Patrick Slade [1] ✉, Mykel J. Kochenderfer [2], Scott L. Delp[1,3] & Steven H. Collins[1]

Physical inactivity is the fourth leading cause of global mortality. Health organizations have requested a tool to objectively measure physical activity. Respirometry and doubly labeled water accurately estimate energy expenditure, but are infeasible for everyday use. Smartwatches are portable, but have significant errors. Existing wearable methods poorly estimate time-varying activity, which comprises 40% of daily steps. Here, we present a Wearable System that estimates metabolic energy expenditure in real-time during common steady-state and time-varying activities with substantially lower error than state-of-the-art methods. We perform experiments to select sensors, collect training data, and validate the Wearable System with new subjects and new conditions for walking, running, stair climbing, and biking. The Wearable System uses inertial measurement units worn on the shank and thigh as they distinguish lower-limb activity better than wrist or trunk kinematics and converge more quickly than physiological signals. When evaluated with a diverse group of new subjects, the Wearable System has a cumulative error of 13% across common activities, significantly less than 42% for a smartwatch and 44% for an activity-specific smartwatch. This approach enables accurate physical activity monitoring which could enable new energy balance systems for weight management or large-scale activity monitoring.

[1] Department of Mechanical Engineering, Stanford University, Stanford, CA, USA. [2] Department of Aeronautics and Astronautics, Stanford University, Stanford, CA, USA. [3] Department of Bioengineering, Stanford University, Stanford, CA, USA. ✉email: patslade@stanford.edu

Effective physical activity monitoring is necessary to understand and overcome inactivity, which is the fourth-largest cause of mortality[1]. Monitoring can relate physical activity to health outcomes, investigate dose–response relationships, and inform health policy. Participating in the recommended amount of physical activity improves musculoskeletal health[2] and weight management[2,3], a pervasive problem in the United States, where 40% of adults are obese[4]. Physical activity impacts perceived quality of life, quality of sleep, and symptoms of depression and anxiety[2,5]. Health policy committees have requested a tool to objectively monitor physical activity at a large scale using a metric like metabolic energy expenditure[2].

Accurately estimating daily energy expenditure requires capturing both basal and active energy expenditure. Basal energy expenditure during basic functions such as breathing comprises a significant portion of daily expenditure and can be estimated with 5–10% error using subject-specific information, such as age and weight[6]. Estimating daily active energy expenditure requires monitoring common and high-expenditure activities such as walking, stair climbing, running, and biking. Walking and stair climbing are necessary for mobility and have the highest energy expenditure among activities of daily living[7]. In the United States, running and biking are the most popular forms of exercise by the number of outings and participation[8].

A method for monitoring daily energy expenditure must meet criteria for large-scale, everyday use[2]. Daily energy expenditure should capture both steady-state activities as well as time-varying activities, such as short bouts of walking[9]. Frequent estimation is necessary to capture short bouts and rapid changes in energy expenditure. Large-scale deployment requires accurate estimation for new subjects, without relying on subject-specific calibration. Design guidelines suggest that wearable medical devices should be low-cost, easy to don and doff, and allow normal motion[10]. Restricting computation requirements to mobile devices is necessary for portability and providing real-time estimates. There are many methods for monitoring activity levels or energy expenditure, but most do not meet these requirements for everyday use.

Self-report surveys and step counts are popular methods of monitoring physical activity on a large scale, but both have significant errors. Surveys categorize weekly activity into time spent exercising at light, moderate, or vigorous-intensity levels[11]. This approximate activity may be used to infer health outcomes based on whether people meet recommended activity levels[2]. Unfortunately, surveys are unreliable and difficult to correct because they have low-to-moderate correlation and inconsistent bias when compared with direct measures of activity[12]. Pedometers and smartphones provide step counts and assume a fixed intensity level of walking to estimate a relative amount of activity. Step count accuracy depends on pedometer type and walking speed[13]. Energy expenditure can be estimated from step counts, weight, and subject-specific respirometry data, but have errors of 24% or more and is limited to monitoring walking[14].

Laboratory-based methods accurately estimate energy expenditure during steady-state activities but are not feasible for everyday use. Respirometry requires minutes of breath-based measurements from expensive and intrusive equipment for steady-state energy expenditure estimation[15,16], which prevents everyday use and causes large errors during time-varying activities. Doubly labeled water provides relatively accurate estimates but costs $200–$300 per use[17], limiting scalability; a single estimate requires 7–14 days[18] and may not offer enough information to relate physical activity and health outcomes. Simulation methods use musculoskeletal models[19,20] or walking mechanics[21] for estimation. These simulations require many sensors, take minutes or hours of computation time per estimate, and are challenging to generalize[22,23]. Laboratory-based methods cannot monitor activity at scale but offer accurate steady-state estimates for training models.

Combining wearable sensors and data-driven methods enables portable and computationally efficient estimation, but many methods rely on subject-specific data to train their models and do not evaluate the accuracy for new subjects. Data-driven methods may use subject-specific information, such as weight and height[24], as well as a variety of wearable sensors including accelerometers and inertial measurement units (IMUs)[25,26], heart rate monitors, or electrocardiography[27,28], electromyography, impedance pneumography[29,30], and various combinations[16,29–31]. These methods have shown a high correlation between sensor data and energy expenditure[32,33] and can accurately evaluate physical fitness[34]. Wearable data-driven methods using subject-specific training data have estimated energy expenditure with relatively low errors of 14–27%[16,28,30]. Unfortunately, methods using subject-specific training data have about twice the expected error when estimating energy expenditure for new subjects[35].

Activity monitors and smartwatches have high errors when estimating energy expenditure for new subjects, possibly because they rely on heart rate, wrist kinematics, or trunk kinematics. Most activity monitor estimates are based on the number of acceleration measurements that reach a threshold each minute[36,37]. Activity monitors evaluated with new subjects have been reported to have 30% error for wrist-worn devices[38] and 27% for hip-worn devices[39], suggesting they do not capture motion related to energy expended by lower-limb muscles. Activity monitors typically only estimate during walking or running because they require significant wrist or pelvis motion, precluding activities like biking. Smartwatches and wearable data-driven models report large errors, from 27% to 93% when evaluated with new subjects[30,31,39], with errors varying across brands. Heart rate and respirometry have a delayed response to changes in energy expenditure, which causes errors at the start of steady-state conditions and during time-varying activities.

We hypothesized that a data-driven method without subject-specific training data that relies only on wearable measurements of lower-limb kinematics segmented by stride, without subject-specific training data, could estimate energy expenditure more accurately than state-of-the-art methods during common activities including walking, running, stair climbing, and biking. Lower-limb kinematics could provide more useful information than heart rate, wrist kinematics, or trunk kinematics because lower-limb activities constitute a larger portion of daily energy expenditure than upper-limb activities[7,8]. Lower-limb kinematics converge more quickly than heart rate or respirometry, enabling estimation during time-varying activities. Measuring lower-limb kinematics requires only a few low-cost and wearable IMUs. During these activities, lower-limb kinematics are periodic and can be segmented by stride. This new modeling approach creates time-invariant inputs appropriate for simple data-driven models such as linear regression, which requires minimal computation. Additional information from the percent stride may improve estimation accuracy compared with existing data-driven methods. To evaluate this hypothesis, we performed experiments to select two sensors worn on one leg from a comprehensive set of existing wearable biomechanics sensors. Next, we collected training data and validated a Wearable System with a diverse group of new subjects during new steady-state and time-varying conditions. We expected our results to inform the development of energy balance systems for weight control and data collection tools for relating physical activity to outcomes like cardiovascular health.

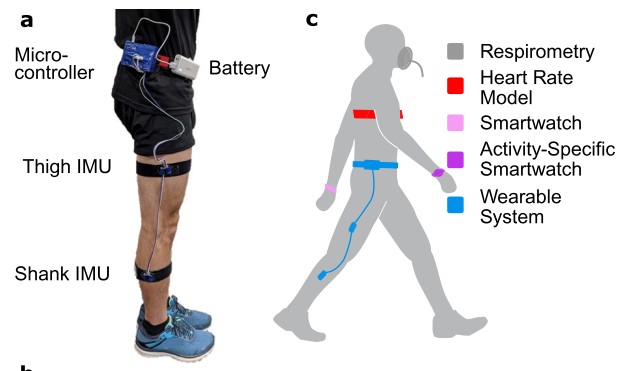

| Activity | Training conditions (*n* = 13) | | | Test conditions (*n* = 24) | |
|---|---|---|---|---|---|
| Walk (m/s) | 0.75 | 1.25 | 1.75 | 1.0 | 1.5 |
| Run (m/s) | 2.25 | 2.75 | 3.25 | 2.5 | 3.0 |
| Stairs (step/min) | 40 | 60 | 80 | 50 | 70 |
| Bike (Watts) | 20 | 80 | 150 | 50 | 120 |

**Fig. 1 The Wearable System components, energy expenditure estimation methods, and experimental conditions. a** The Wearable System consists of a microcontroller, battery, and an inertial measurement unit (IMU) attached to the shank and thigh on one leg. **b** The Wearable System estimated energy expenditure in real-time, using a model trained with data from 13 subjects completing three conditions of walking, running, stair climbing, and biking. The trained model was held constant to test the Wearable System with 24 new subjects during two different steady-state conditions for the same activities and four new time-varying conditions. These subjects had not participated in previous experiments and were selected to emulate a diverse adult population. **c** The Wearable System was compared with commercial devices including the Smartwatch and the Activity-Specific Smartwatch as well as baselines including the Heart Rate Model, Per-Breath Respirometry, and Fast-Estimated Respirometry.

## Results

**Evaluating the wearable data-driven method.** In order to monitor physical activity by accurately estimating energy expenditure, we evaluated several methods, explored sensor importance, and extended estimation to time-varying conditions. We estimated energy expenditure during steady-state activities where each healthy subject ($n = 13$) participated in one condition of sideways walking, backward walking, and hopping as well as multiple conditions of normal walking, loaded walking, and running (Supplementary Fig. 1a). We measured ground truth energy expenditure by averaging respirometry measurements from the last 3 mins of each condition, which we refer to as Steady-State Respirometry. An initial exploration estimated energy expenditure during time-varying conditions where healthy subjects ($n = 4$) transitioned back and forth between walking and running with a 30 second period. We approximated ground truth during time-varying conditions by interpolating between subject-specific Steady-State Respirometry values from walking and running conditions based on treadmill speed, which we refer to as Interpolated Respirometry. The data were recorded with tethered respirometry and wearable sensors including electromyography, force-sensing insoles, IMUs, and a heart rate monitor (Supplementary Fig. 1b). We estimated metabolic energy expenditure with several methods including the Heart Rate Model, the Activity Monitor, the Musculoskeletal Model using muscle-level energy estimates, the Data-Driven Model using all wearable sensor data segmented by stride in a linear regression model (Supplementary Fig. 2), Per-Breath Respirometry, and Fast-Estimated Respirometry which fit laboratory-based respirometry measurements to a first-order exponential function for quicker steady-state estimates. Accuracy was evaluated by removing one

subject and one condition from the training data, fitting a model to the training data, estimating energy expenditure for the withheld subject and condition, and comparing it to ground truth. We did this for all permutations of subjects and conditions and averaged the error.

We validated that the Data-Driven Model using all wearable sensor inputs was the best wearable method for estimating energy expenditure because it had the lowest errors during steady-state and time-varying conditions. The Data-Driven Model had a 10.5% relative error during steady-state conditions, about half the error of the second-most accurate model (Supplementary Fig. 3a). The Data-Driven Model estimates were constant from the start of the condition because input signals converged quickly, as expected (Supplementary Fig. 3b). Fast-Estimated Respirometry had the lowest steady-state error after 74 seconds, confirming its usefulness as a laboratory-based test where accuracy is paramount and longer trial times are acceptable. During time-varying conditions, the Data-Driven Model had a 7% absolute error, about one-quarter the error of the next-most accurate model (Supplementary Fig. 4). Physiological signals had a delayed response to changing energy expenditure, causing significant error in steady-state and time-varying conditions.

The Data-Driven Model error decreased with additional training data (Supplementary Fig. 5) and increased when estimating activities with energy expenditure values significantly different from the training data. Holding out and evaluating conditions other than running resulted in errors between 10% and 15%, indicating some generalization to new conditions. Holding out and evaluating running conditions resulted in the largest errors because the model estimated smaller energy expenditure values, similar to the training data (Supplementary Fig. 6). Thus, the similarity of training data impacts the error when estimating new conditions. The approach used by the Data-Driven Model requires training data for all activities that will be monitored. The Musculoskeletal Model was less sensitive to the choice of training data. Combining these two methods may perform best when estimating energy expenditure for many new activities. We focused on estimating known activities and selected the Data-Driven Model because it had greater accuracy and required minimal computation when trained on sufficient data.

**Selecting informative sensors.** The Data-Driven Model was evaluated with different inputs to select a simple and informative set, and an IMU on the shank and thigh of one leg was found to have the lowest steady-state error. We first compared inputs from permutations of four sensor classes including inertial measurements, muscle activity, kinematics, and vertical ground reaction forces. We evaluated these existing sensors, rather than develop new sensors, to ensure the needed sensors would be commercially available. The IMUs were selected because they had the lowest error when using only one sensor class (Supplementary Table 1). We then compared all permutations of the IMUs, each of which had a triaxial accelerometer and gyroscope (Supplementary Table 2). The best results were achieved with one sensor on the shank and one on the thigh. Input from these two sensors had a 13.7% relative error, compared with 10.5% error when using inputs from all sensors. Using a single IMU on the thigh resulted in an error of 16.7%, indicating a single sensor such as a smartphone strapped to the leg may not be quite as accurate.

The selected IMUs offered some additional benefits beyond other wearable sensors. They are low-cost, which may enable large-scale use. They are lightweight, compact, easy to don and doff, and allow uninhibited motion. Processing data from these sensors required little computation; a portable microcontroller

computed real-time energy expenditure estimates in 0.01 s (Supplementary Table 3).

**Designing and training the Wearable System**. Next, we built the Wearable System, which consisted of a microcontroller and battery worn on a belt as well as two IMUs worn on the shank and thigh of one leg (Fig. 1a and Supplementary Movie 1). The total system cost was approximately $100 in retail components. The Wearable System weighed 232 grams and could estimate energy expenditure for 7.3 h on a single charge.

The Wearable System estimated energy expenditure using a new Data-Driven Model trained with a combination of previously collected and new data. The previously collected walking and running data used to select the best estimation method were combined with a new experiment where healthy subjects climbed stairs and biked ($n = 11$) (Fig. 1b). The stair climbing conditions were collected on a stairmill at speeds of 40, 60, and 80 steps per minute. The biking conditions were collected on a stationary bike at resistance levels of 20, 70, and 150 Watts while pedaling at 80 revolutions per minute. Pilot tests revealed that changes in the placement of the IMUs resulted in large estimation errors. To address this, we applied random rotations to the sensors in the training data to create synthetic data. The Data-Driven Model was trained with this synthetic data to improve robustness to orientation errors. The Wearable System used this robust model for all evaluations with new subjects and new conditions.

**Evaluating the Wearable System with new subjects and new conditions**. We compared the Wearable System to commercial products and other methods of estimating energy expenditure to determine the estimation error for new subjects and new conditions. The methods we compared included the Smartwatch, the Activity-Specific Smartwatch, the Heart Rate Model, the Activity-Specific Model, Per-Breath Respirometry, and Fast-Estimated Respirometry (Fig. 1c). The methods were evaluated with a group of diverse subjects ($n = 24$, 15 men and 9 women; age = 34.8 ± 11.6 years; body mass = 74.3 ± 13.1 kg; height = 1.73 ± 0.07 m; body mass index = 24.9 ± 4.1) that had not participated in previous experiments. The subjects completed two new steady-state conditions at intermediate speeds between the three training conditions collected for each activity, as well as four new time-varying conditions. The steady-state conditions included walking at 1.0 and 1.5 m/s, running at 2.5 and 3.0 m/s, climbing stairs at 50 and 70 steps per minute, and biking with a resistance of 50 and 120 Watts. The ground truth energy expenditure during steady-state conditions was Steady-State Respirometry. The time-varying conditions periodically changed treadmill speed for discrete steps between quiet standing and walking, sinusoidal walking speeds, discrete steps between walking and running, and sinusoidal walking and running speeds. Interpolated Respirometry approximated ground truth during time-varying conditions by interpolating between subject-specific Steady-State Respirometry values from walking and running conditions based on treadmill speed.

The Wearable System used inputs of lower-limb kinematics which converged quickly from the start of steady-state conditions, resulting in consistent, low-error estimates (Fig. 2a). The other methods relied on physiological signals that had delayed responses to changing energy expenditure and slowly reached steady state. The other methods had steady-state errors that varied widely by activity (Supplementary Fig. 7). The Wearable System had the lowest error for the first 44 s, after which the laboratory-based, Fast-Estimated Respirometry had the lowest error (Fig. 2b). Although respirometry is not feasible for large-scale monitoring, it offers accurate steady-state estimates for

training models. the Heart Rate Model, the Smartwatch, and the Activity-Specific Model rely on physiological signals and have large initial errors, which decrease over time, but still result in large steady-state errors.

The Wearable System had the lowest steady-state errors of the wearable methods, closely matching Steady-State Respirometry (Supplementary Movie 2). The steady-state error of the Wearable System was lower than all other methods (paired $t$ tests: $p \leq 1 \times 10^{-6}$). During steady-state conditions, the Wearable System had 13% steady-state error, about half the error of the second-most accurate model, the Activity-Specific Smartwatch (Fig. 2c). The steady-state errors for the Smartwatch and the Activity-Specific Smartwatch match those from previous studies[31,39]. Even the Activity-Specific Model, which used manual labeling during steady-state conditions to achieve perfect activity classification, had higher steady-state error than the Wearable System (18%). The cumulative energy expenditure error of the Wearable System was 12%, significantly lower than with the other methods, which had errors ranging from 38% to 71% (paired $t$ tests: $p \leq 2 \times 10^{-14}$) (Fig. 2d).

The Wearable System accurately estimated energy expenditure during time-varying conditions (Supplementary Movie 3). The Wearable System captured changes in energy expenditure while the other models had incorrect and delayed estimates (Fig. 3a–d). The error over time for the Wearable System was 23%, significantly less than 46–105% error for other methods (paired $t$ tests: $p \leq 3 \times 10^{-4}$) (Fig. 3e). The cumulative energy expenditure error followed a similar trend (paired $t$ tests: $p \leq 7 \times 10^{-4}$) (Fig. 3f). Across all steady-state and time-varying conditions, the Wearable System had a cumulative energy expenditure error of 13%, significantly less than 42–86% for other methods (paired $t$ tests: $p \leq 2 \times 10^{-21}$). We also evaluated a version of the Wearable System using only data from the thigh IMU and found a cumulative error of 19%.

We visualized the linear regression model used by the Wearable System to understand how conditions were differentiated to achieve low error. The magnitude of the model weight assigned to an input indicated importance (Fig. 4a). Larger weights were assigned to the gyroscope inputs. Certain portions of each stride also had larger weights, such as the second and fifth quintiles. The model weighting as a function of percent stride was similar to the standard deviations of input signals across activities. The normalized dot product between the weights and standard deviation of input signals resulted in a cosine similarity of 0.96, indicating the high-dimensional trends were similar. We plotted the most informative input signal alongside the model weights to illustrate how it contributed to an estimate of energy expenditure (Fig. 4b). Interpretation of how inputs contribute to estimates is difficult as all inputs are compared at once.

Subject surveys found the Wearable System to be comfortable and have high usability. Twenty-one subjects who participated in the validation of the Wearable System were surveyed. The usability was evaluated with the System Usability Scale[40]. The Wearable System had a relatively high overall score of 80.9 out of 100 averaged across subjects (Supplementary Table 4), which is the 90th percentile among 5000 device surveys that used the System Usability Scale[41]. Comfort-related metrics were evaluated with a survey based on the Questionnaire for User Interaction Satisfaction[42]. The Wearable System had high scores associated with different comfort-related attributes (Supplementary Table 5). This indicates the Wearable System, though only a research prototype, has the potential to be adapted for use in clinical or at-home settings.

## Discussion

The approach used in the Wearable System meets the requirements for devices that monitor physical activity on a large scale,

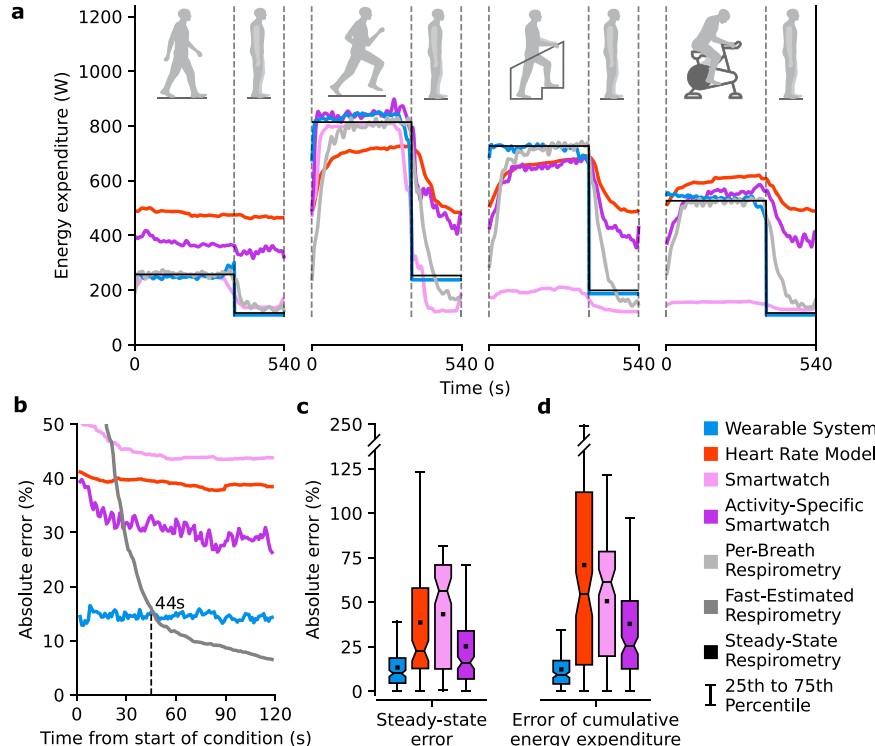

**Fig. 2 Estimating energy expenditure during steady-state conditions. a** Methods estimating energy expenditure were evaluated during 6-minute steady-state conditions followed by 3 mins of quiet standing. Two conditions were collected for each activity in a randomized order. The estimates were interpolated at 1 s intervals, averaged over subjects, and low-pass filtered. **b** The error as a function of time from the start of the condition evaluated how quickly the methods reached a steady-state estimate, averaged across all conditions. The absolute percent error was computed by taking the mean absolute error between the estimates and ground truth Steady-State Respirometry. **c** The steady-state error represents the mean absolute percent error between the estimates of each method averaged over the last 3 mins of a condition and Steady-State Respirometry ($n = 24$ subjects). Per-Breath Respirometry and Fast-Estimated Respirometry were not compared because they become Steady-State Respirometry when averaged over multiple minutes. **d** Cumulative energy expenditure is the total expenditure during each steady-state condition, including expenditure above quiet standing during the 3 mins following the condition ($n = 24$ subjects). The cumulative error represents the mean absolute percent error between cumulative estimates for each method and Per-Breath Respirometry. The boxes extend from the lower to upper quartile values of the data, with a line at the median and a dot at the mean. The whiskers extend to the last data point within 2.5 times the interquartile range. Fliers were not plotted due to the wide range of errors.

thereby providing a tool that could be used in weight management systems and for studies that relate physical activity to health outcomes. The Wearable System used low-cost IMUs worn on the shank and thigh of one leg to estimate energy expenditure in real-time (Supplementary Movie 1). This approach is portable and unobtrusive, meeting the criteria to allow for everyday use. We demonstrated that the Wearable System accurately estimated energy expenditure during steady-state and time-varying activities, including walking, running, stair climbing, and biking, the activities that contribute the most to daily energy expenditure. The Wearable System was evaluated with a diverse group of subjects, indicating it may effectively monitor most adults. The Wearable System estimated energy expenditure once per stride, possibly enabling new types of studies, such as characterizing the health benefits of acute dose–response with repeated, low-intensity exercise[11]. The cumulative energy expenditure error of the Wearable System was one-third the error of state-of-the-art wearable methods, a substantial improvement that could better relate physical activity to health outcomes.

The Activity Monitor had twice the steady-state error of the Data-Driven Model, which may be because it relied on pelvis kinematics, rather than lower-limb kinematics, and did not incorporate stride information. The success of the data-driven approach seems to be largely attributable to the selection of lower-limb kinematics as inputs and segmenting data by stride. A version of the Data-Driven Model using only trunk kinematics as inputs resulted in twice the steady-state error, reinforcing the finding that using lower-limb kinematics was beneficial (Supplementary Fig. 3a). When the Activity Monitor and Data-Driven Model were given the same trunk kinematics as inputs, estimation differences were due to differences in data segmentation. The Data-Driven Model segmented data by stride for time-invariant estimates and achieved lower error than the Activity Monitor, which counted acceleration measurements that reached a threshold. Future devices designed to estimate energy expenditure may benefit from using sensors that measure lower-limb kinematics and models that exploit stride structure.

The Activity-Specific Smartwatch and the Smartwatch had a threefold higher cumulative error than the Wearable System, potentially because of a reliance on heart rate and wrist kinematics rather than lower-limb kinematics. Previous Smartwatch studies report similar errors from 35% to 93% when estimating energy expenditure for new subjects[30,31,39], supporting the idea that Smartwatches have significant errors. The sensor inputs and models used by the Smartwatches are not publicly available, but our results indicate that the Activity-Specific Smartwatch relies on heart rate and the Smartwatch relies on wrist kinematics (Supplementary Fig. 7). The Activity-Specific Smartwatch exhibited similar trends to the Heart Rate Model, including higher estimates during walking and quiet standing, estimates drifting during conditions, and a delayed response to changes in energy expenditure (Fig. 2a). The Smartwatch estimates during stair

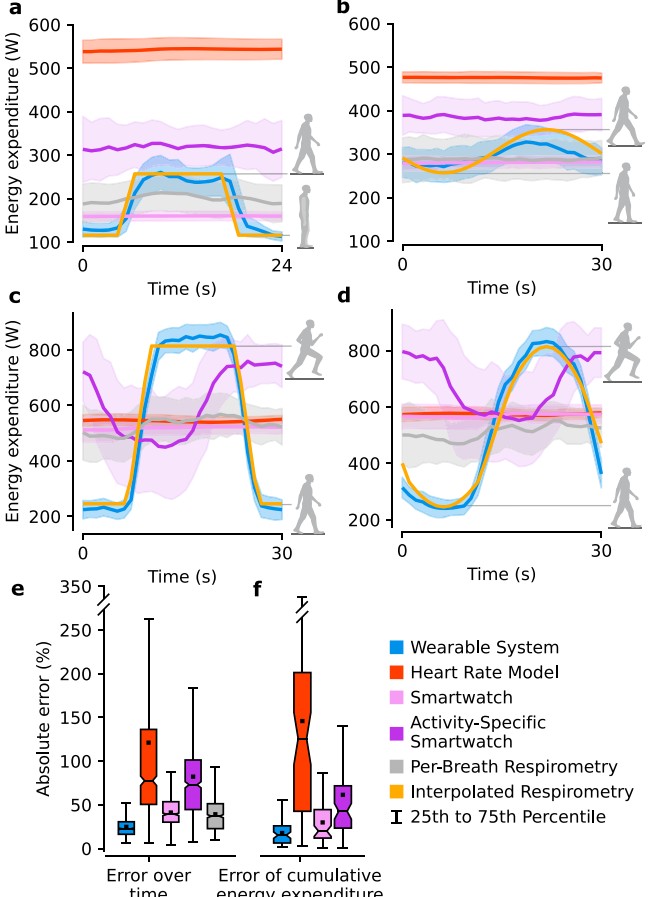

**Fig. 3 Estimating energy expenditure during time-varying conditions.** The time-varying conditions consisted of periodic profiles of treadmill speeds including **a** discrete steps between quiet standing and walking, **b** sinusoidally varying walking, **c** discrete steps from walk to run, and **d** sinusoidally varying walk to run. The mean and standard deviation of each estimation method is represented as a line with error bands. Interpolated Respirometry approximated ground truth by interpolating between subject-specific Steady-State Respirometry values from walking and running conditions based on treadmill speed. The error profiles represent one standard deviation. **e** The error over time represents the mean absolute percent error between estimates from each method and the Interpolated Respirometry ($n = 24$ subjects). **f** Cumulative energy expenditure error was computed as the mean absolute percent error between cumulative estimates for each method and Per-Breath Respirometry ($n = 24$ subjects). The boxes extend from the lower to upper quartile values of the data, with a line at the median and a dot at the mean. The whiskers extend to the last data point within 2.5 times the interquartile range. Fliers were not plotted due to the wide range of errors.

climbing and biking were substantially lower than ground truth values, possibly because subjects held onto the handrails or handlebars, which minimized wrist motion (Supplementary Fig. 7). The Smartwatches may have higher error because they do not seem to segment data by stride, as indicated by the large initial steady-state error and drift in estimates. The Wearable System had a consistent level of error across all steady-state conditions (Supplementary Fig. 7). Previous experiments found activity monitors to estimate energy expenditure with 30% error for wrist-worn devices[38] and 27% for hip-worn devices[39], suggesting that these locations do not capture motion related to energy expended by lower-limb muscles as accurately as sensors

placed on the legs. These results suggest the wrist is a suboptimal placement for sensors when estimating energy expenditure.

The finding that the Wearable System provides significantly more accurate energy expenditure estimates than existing wearable methods by using two carefully selected sensors is surprising. From an information-theoretic perspective, we would expect that selecting multiple sensors would achieve at least the same accuracy as a single prescribed sensor, such as a Smartwatch. However, the careful selection of two IMU sensors enabled the Wearable System to have three times lower cumulative error than the Smartwatches. A version with inputs from the single-thigh IMU still had less than half the cumulative error of the Smartwatches. Including heart rate, a signal used in many wearable methods for estimating energy expenditure, as an input to the Wearable System did not improve the accuracy. Even using comprehensive sensor measurements of leg kinematics and major muscle groups performed similarly to the two selected IMU sensors. The Wearable System was more accurate than wearable data-driven methods using a variety of sensors and subject-specific training data with errors of 14–27%[16,28,30], which would have approximately twice the error when evaluating new subjects[35]. When designing wearable devices, rigorous sensor selection may provide counterintuitive results that can significantly improve performance.

The Smartwatches' large errors were likely due to a reliance on heart rate, rather than a difference in training data. The Smartwatch and Heart Rate Model had similar average steady-state errors of 43% and 39% and match trends across individual conditions (Supplementary Fig. 7). The Heart Rate Model had significantly higher error than the Wearable System, despite being trained with the same data. This suggests that heart rate was less informative than lower-limb kinematics. No information is available about the data used to train the commercial Smartwatch methods. The similarity between the Heart Rate Model and commercial Smartwatch methods indicates they would likely have significantly higher error than the Wearable System, even if trained from the same data. The Wearable System estimates had no statistically significant correlation between error and age, height, weight, or body mass index. The results from evaluating the different methods with a diverse group of subjects suggest the Wearable System would likely be more accurate than the Smartwatch methods when monitoring most adults.

The Wearable System estimated time-varying energy expenditure much more accurately than other methods because it was based on lower-limb kinematics that more closely tracked the activity in the leg muscles, which contribute the most to energy expenditure. The Wearable System estimates were accurate from the start of steady-state conditions (Fig. 2b and Supplementary Movie 2), captured sinusoidally (Fig. 3b, d), and step (Fig. 3a, c) changes in energy expenditure during time-varying conditions, and had a low cumulative error. Lower-limb kinematics are known to quickly converge to steady-state following gait changes and track continuous fluctuations in walking speed. The other methods relied on physiological signals which had a delayed response to changing energy expenditure that caused significant error during the start of steady-state conditions, throughout time-varying conditions, and cumulatively (Fig. 3e, f). We estimated instantaneous energy expenditure during time-varying conditions using measured energy expenditure from steady-state conditions and interpolation based on speed. This assumes the instantaneous energy expenditure matches the work rate of the subject at each instant in time, which has been shown to be a good assumption for walking conditions below the aerobic threshold[43]. The time-varying conditions had a period of 30 seconds, faster than the delays in breath-to-breath measurements. This 30-second period was selected to minimize energy expenditure associated with

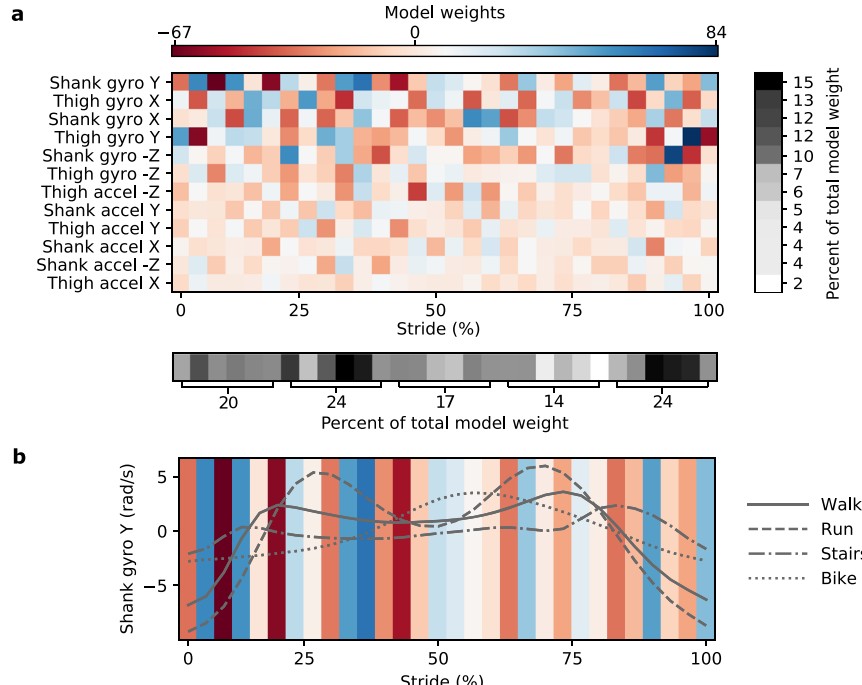

**Fig. 4 Interpreting the Wearable System. a** The Wearable System used a linear regression model to estimate energy expenditure with inputs from inertial measurement units worn on the shank and thigh. The inertial measurement units consist of a three-axis accelerometer (accel) and a three-axis gyroscope (gyro). The X, Y, and Z directions represent the fore-aft, mediolateral, and vertical axes, respectively. These orientations are assigned during quiet standing, but the sensors are attached to different body parts that move relative to the global reference frame. Input data were low-pass filtered, split by stride, and discretized each stride to a fixed input size before being used in the model. The model used ridge regularization that penalized the sum of the squared model weights. A larger magnitude weight indicated a more informative input. The input signals are shown in descending order of importance based on contribution to total model weight. Gyroscope inputs were more informative than accelerations. Inputs in the second and fifth quintile of each stride were the most informative. **b** The input signal with the largest weight, the gyroscope measurement of the shank in the sagittal plane, was plotted for each activity alongside the model weights. The linear regression model may relate the absolute value of signals to energy expenditure or differentiate two points of the input data to approximate a derivative. Interpretation is challenging as all inputs are compared at once.

changing speeds by maintaining a low average acceleration of 0.07 m/s². Prior experiments found that varying walking speeds by 0.6 meters per second sinusoidally, with a 4-second period (0.15 m/s² acceleration), increased energy expenditure by 4–8%[44]. Assuming that increased energy cost is linearly related to acceleration, the cumulative energy expenditure during time-varying conditions would be ~2–4% higher than the interpolated estimates of instantaneous energy expenditure, because the interpolation between steady-state energy expenditure values does not include the costs of additional acceleration.

Visualizing the linear regression model used by the Wearable System revealed that gyroscope measurements and percent stride of input data contributed to accurate estimation. The linear regression model had larger weights assigned to gyroscope inputs (Fig. 4a), possibly because angular velocity better captured the motion of two bodies connected by a pin joint. Model weight varied across each stride with large weights during the second and fifth quintiles, indicating knowledge of phase within a stride is important for estimation. Portions of each stride with larger weights also had larger standard deviations in input signals across activities. The model may use phases with large differences to distinguish activities, similar to activity classification. The Wearable System still had lower steady-state error than the Activity-Specific Model even given the fact that the Activity-Specific Model was based on ideal activity classification performed manually. This indicates that the Wearable System may extract additional information, such as how the activity was performed, where changes in motion may relate to muscle-level energy expenditure.

Interpreting how the Wearable System estimates energy expenditure from input signals is challenging because all inputs are compared at once. Simple, interpretable mechanisms by which the linear regression model might operate include relating the absolute value of signals to energy expenditure or differentiating two points of the input data to approximate a derivative. To illustrate this, we compare the most informative input signal, the gyroscope measurement of the shank in the sagittal plane, alongside the model weights (Fig. 4b). The 10th and 11th points in the stride have a large positive weight and the running input signal has the highest values at these points; the model may directly associate this signal with large energy expenditure during running. The 3rd and 4th points in the stride have large weights, one positive and one negative, which may differentiate these two points and distinguish the walking condition, which has a large positive slope, from biking, which has a slope near zero. Segmenting input signals by stride creates 360 parameters to capture information for accurate estimation, but a simultaneous comparison of so many inputs makes exact model interpretation impossible. Although we cannot say for sure, the model seems to utilize absolute weighting and differentiation to distinguish conditions.

The approach used by the Wearable System may be effective for monitoring physical activity, but improved hardware would be necessary for large-scale deployment. The Wearable System was a proof-of-concept device with a bulky microcontroller and wired sensors. Despite being a prototype, the subject surveys reported that the Wearable System had a high usability rating (Supplementary Table 4) and high ratings in metrics rated to

comfort (Supplementary Table 5). This is likely owing to the light weight and small form factor of the IMUs. A demonstration of donning and doffing the device shows that these are simple procedures, taking 25 and 14 seconds, respectively (Supplementary Movie 4). A future product would need to be refined by using common sensors such as a smartphone worn on the thigh, IMUs embedded into a shoe or integrated into clothing, or stickers powered by heat or sweat[45]. A smartphone could provide wireless sensor communication and compute estimates. This future product will also require investigation to determine if it is convenient enough for large-scale, everyday use.

The Wearable System accurately estimated energy expenditure for the tested conditions, but additional data collection may improve performance and the activities that can accurately be monitored. The Wearable System was trained with data from 13 young and healthy subjects during four common activities. Collecting data from a larger and more diverse group of subjects and activities would create a more accurate and general model, which may require additional information such as subject age. Additional training data would need to be collected and used to retrain the Wearable System to accurately estimate energy expenditure for any other activities. Early evaluations of other machine learning models found worse performance than linear regression, likely because they overfit the relatively small amount of training data. Future studies with access to more data may benefit from a more expressive model, such as a neural network. Activities that have significant upper-limb activity may be challenging to detect without additional sensors but are not among the most common activities with high energy expenditure and thus may comprise only a small portion of active energy expenditure[7,8]. Likewise, activities beyond the aerobic threshold were not considered owing to the challenges of obtaining ground truth energy expenditure values. Accurately estimating conditions with similar kinematics but different energy expenditures, such as biking at different resistances, could be susceptible to error if people were to consistently use the same kinematics. However, in freely selected behaviors, people tend to have large variations in kinetics and muscle activity[46], and we did not observe this issue in the fixed-cadence cycling trials in this study. Evaluating other possible signals, such as respiration frequency[47], might help differentiate energy expenditure for conditions with similar kinematics.

The approach demonstrated by the Wearable System may offer a solution to one of the challenges facing engineers, physicians, and global health organizations: accurately and objectively monitoring common physical activities with a portable, low-cost system. The accuracy when evaluated with a diverse population indicates the Wearable System may be ready to be deployed for effective physical activity monitoring of most adults. The Wearable System relies on two low-cost sensors that may simplify deployment at a large scale, especially for low-income countries and countries with high mortality rates from inactivity. Clinical researchers studying obesity and physical activity could use systems like this one to more accurately investigate how the activity relates to health outcomes and inform health policies. Engineers designing wearable devices could leverage our sensor selection approach to improve the effectiveness of their devices. Machine learning researchers may find our insights into using gait cycle structure to create time-variant inputs helpful for modeling other challenges related to motion and physiology. Biomechanics researchers may be interested in the implications of the selected sensor signals and locations for monitoring and understanding gait and energy expenditure. The Wearable System could be combined with a method for estimating caloric intake to create an energy balance system. Monitoring and understanding energy balance could enable personalized weight management tools to reduce obesity.

## Methods

**Experimental design**. The research objective was to compare errors when estimating energy expenditure of the Wearable System method and state-of-the-art wearable methods. We hypothesized that the Wearable System would have significantly lower error than the state-of-the-art methods. The necessary sample size to validate the Wearable System was found to be 15 subjects from a power analysis based on earlier experiments that estimated the Wearable System would have 14% absolute error, a standard deviation of 12%, a difference of at least 14% with each compared method, and a power of 0.9. We stopped data collection after reaching 25 subjects in case of sensor failures and because some subjects were not able to complete all conditions. One subject was excluded because of sensor failure and no other exclusions were made. All subjects were volunteers and provided written informed consent before completing the protocol IRB-17282 approved by the Stanford University Institutional Review Board. The authors affirm that human research participants provided informed consent for publication of Supplementary Movies 1–4. The experiments consisted of human subject testing in a laboratory experiment, where each subject performed the same set of conditions in a randomized order. Each of the four experiments performed in this study is detailed in the next four sections. These sections each describe the motivation for the experiment, which sensors were used to collect data, and how the data were collected. The subheading for each experiment matches the corresponding subheading in the Results section.

**Evaluating the wearable data-driven method and selecting informative sensors**. In order to evaluate the data-driven method and select the fewest sensors necessary to estimate energy expenditure, we performed an experiment to collect data from a variety of conditions and wearable sensors for offline testing. Healthy young adults ($n = 13$, 8 men and 5 women; age = $23.8 \pm 2.6$ yr; body mass = $68.3 \pm 10.6$ kg; height = $1.72 \pm 0.07$ m) completed 12 steady-state conditions on a treadmill (Supplementary Fig. 1a). One subject was excluded owing to sensor failure. The conditions were completed in the order: quiet standing, six speeds of walking and running between 0.75 and 3.25 m/s, sideways walking at 1 m/s, backwards walking at 1 m/s, hopping in place at a self-selected rate and height, and loaded walking at 1.25 m/s with an additional 10% and 20% of the subject's bodyweight. The conditions lasted 5 mins followed by quiet standing for at least 1 minute. The minimum, maximum, and average energy expenditures across conditions were 179, 1295, and 471 Watts.

Biomechanics data were collected with a variety of wearable sensors (Supplementary Fig. 1b). Tethered respirometry equipment (Quark CPET, COSMED) measured the volume of carbon dioxide and oxygen exchanged per breath. A heart rate monitor (Dual Heart Rate Monitor, Garmin) recorded heart rate in beats per minute at the same intervals as the respirometry measurements, for the last nine subjects. Electromyography sensors (Trigno IM, Delsys Inc.) recorded muscle activity at 2000 Hz from seven muscles of the left leg: soleus, medial and lateral gastrocnemii, tibialis anterior, vastus medialis, rectus femoris, and biceps femoris. IMUs (MTw Awinda, Xsens) recorded motion at 100 Hz from seven sensors placed on the pelvis and the foot, shank, and thigh of both legs. Force-sensing insoles (Pedar, Novel) in both shoes recorded vertical ground reaction forces at 50 Hz. The data processing steps for these sensors are detailed in the "Sensor data processing" subsection. The wearable data were used to compare a variety of methods including Steady-State Respirometry, Fast-Estimated Respirometry, the Activity Monitor, the Activity-Specific Model, the Data-Driven Model, and the Musculoskeletal Model.

**Extending estimation to time-varying conditions**. The second experiment collected the same sensor data and steady-state conditions with additional time-varying conditions that periodically transitioned between walking and running. Healthy young adults ($n = 4$, 2 men and 2 women; age = $23.8 \pm 2.6$ yr; body mass = $67.1 \pm 10.7$ kg; height = $1.71 \pm 0.07$ m) completed the protocol from the first experiment and two time-varying conditions where the speed of walking and running periodically followed a sinusoid or four discrete step changes between 1.25 m/s and 2.75 m/s. The sinusoidal change in speed had a period of 30 seconds. The discrete steps were speeds of 1.25, 1.75, 2.25, and 2.75 m/s. Each discrete step change had a 2 s acceleration followed by eight seconds of a constant speed, resulting in a total period of 60 seconds. The time-varying conditions started with 10 seconds of walking at 1.25 m/s and then periodically changed the speed for 5 mins.

The instantaneous energy expenditure during time-varying conditions cannot be directly measured with respirometry due to a delayed response to changing energy expenditure. Interpolated Respirometry approximated ground truth by interpolating between subject-specific Steady-State Respirometry values from walking and running conditions based on treadmill speed. The methods for estimating time-varying energy expenditure included: the Data-Driven Model, the Heart Rate Model, Per-Breath Respirometry, and Fast-Estimated Respirometry.

**Designing and training the Wearable System**. The third experiment combined a new collection of stair climbing and biking conditions with the previous walking and running data from the first experiment to train the Data-Driven Model used in the Wearable System. Healthy young adults ($n = 10$, 5 men and 5 women;

age = 24.5 ± 2.5 yr; body mass = 65.5 ± 11.0 kg; height = 1.72 ± 0.07 m) completed three steady-state conditions for both stair climbing and biking. The speeds for stair climbing were 40, 60, and 80 steps per minute, which were selected to center around the speed for healthy older adults. A stairmill was used to emulate continuously climbing stairs while remaining in one location for data recording. The kinematics and muscle activity for climbing stairs and using a stairmill follow similar trends. The biking conditions had resistance levels of approximately 20, 70, and 150 Watts. The subject manually followed a pedal rate of 80 revolutions per minute based on findings that this is a comfortable rate[48].

The Wearable System was assembled from a few off-the-shelf components and estimated energy expenditure in real-time with the Data-Driven Model trained from experimental data (Supplementary Fig. 2a). The Wearable System consisted of a Raspberry Pi 3b+, a rechargeable battery, and two Adafruit Precision NXP Breakout Boards. Computation and data storage were performed on the Raspberry Pi in real-time. The Data-Driven Model relied on inputs from the two IMUs worn on the shank and thigh of the left leg.

The Wearable System identified quiet standing when strides were not detected for at least 8 seconds and used a heuristic to estimate energy expenditure. If a stride was not detected for 8 seconds, the Wearable System switched to estimate quiet standing from a scaled estimate of basal energy expenditure. A previous equation was used to estimate basal energy expenditure from a subject's height, weight, age, and gender[6]. The scaling factor was computed by dividing Steady-State Respirometry estimates during quiet standing by basal estimates and averaging across all training subjects. When evaluating the energy expenditure during quiet standing for a new subject, their basal estimate was multiplied by this scaling factor.

**Evaluating the Wearable System with new subjects and new conditions**. The fourth experiment compared the Wearable System and state-of-the-art wearable methods when estimating energy expenditure for new subjects and new steady-state conditions during walking, running, stair climbing, and biking as well as four time-varying activities. The subjects were a diverse group of adults that had not participated in any previous experiments (n = 24, 15 men and 9 women; age = 34.8 ± 11.6 yr; body mass = 74.3 ± 13.1 kg; height = 1.73 ± 0.07 m; body mass index = 24.9 ± 4.1). We selected participants to achieve similar mean and standard deviation values of the age, body mass, height, and body mass index of a previously diverse study validating the error of Smartwatches estimating energy expenditure[39]. The subjects could skip any conditions they were unable to complete. The new conditions were selected at intermediate speeds or resistance levels between the conditions collected to train the Wearable System. The steady-state conditions included walking at 1.0 (n = 24) and 1.5 (n = 24) m/s, running at 2.5 (n = 13) and 3.0 (n = 11) m/s, climbing stairs at 50 (n = 22) and 70 (n = 14) steps per minute, and biking with a resistance of 50 (n = 24) and 120 (n = 24) Watts while pedaling at 80 revolutions per minute. The n values represent the number of the 24 subjects that completed each condition. The time-varying conditions consisted of periodic profiles of varying treadmill speeds including discrete steps between quiet standing and walking at 1.0 m/s, sinusoidally varying walking speed between 1.0 and 1.5 m/s, discrete steps from walking at 1.0 m/s to running at 3.0 m/s, and sinusoidally varying from walking at 1.0 m/s to running at 3.0 m/s. Each condition lasted 6 mins followed by at least 3 mins of quiet standing. The conditions were collected in two sections, one consisting of walking and running conditions and the other of stair climbing and biking because the cart containing the respirometry equipment had to be moved to reach the exercise equipment. The order of the two sections and the order of the conditions within each section were randomized. The evaluated methods included: the Wearable System, the Smartwatch, the Activity-Specific Smartwatch, the Heart Rate Model, Per-Breath Respirometry, Fast-Estimated Respirometry, and Steady-State Respirometry.

A post hoc survey of 21 subjects was used to evaluate the usability of the Wearable System as well as metrics related to comfort. The usability questionnaire was the standard System Usability Scale[40], which is a Likert scale meant to evaluate the usability of a system. Twenty-one participants evaluated the Wearable System with this survey. The metrics related to comfort were based on the Questionnaire for User Interaction Satisfaction survey[42].

**Sensor data processing**. Wearable sensor data from the first two experiments were processed before being input to any models. Electromyography signals were filtered with a 30–500 Hz bandpass filter, rectified, filtered with a 6 Hz low-pass filter, and normalized by the maximum signal for each muscle during walking at 1.25 m/s. The filters were fourth-order, bidirectional Butterworth filters. Some measurements from the force-sensing insoles reported negative forces, which were corrected by replacing these values with the first preceding non-negative measurement. The force measurements also drifted over time and were corrected by shifting the minimum value of each 10-second window of data to 0, approximating having no vertical ground reaction force during the flight phase of each stride.

The sagittal kinematics for the ankle, knee, and hip angles were computed by passing orientation data from the IMUs to OpenSense, an open-source tool relying on OpenSim musculoskeletal simulation[19]. OpenSense assumed the subject was standing to calibrate a human Musculoskeletal Model. The IMUs provided data to an inverse kinematics solver to estimate the lower-limb kinematics while meeting musculoskeletal constraints. The initial calibration and solver were reset for each condition to limit the error accumulated from sensor drift. The computed joint

kinematics from OpenSense were compared to those from a motion capture system (Optitrack), resulting in an absolute error of five degrees averaged across all sagittal joints and conditions.

Sensor data were synced with timestamps from each device and segmented by stride. The first two experiments resampled all sensor data to 100 Hz and detecting strides by detecting when the pressure insole of the right foot reported in increasing force that crossed a threshold of 150 N. The Wearable System filtered and processed inertial measurement data and used sagittal plane measurements to segment strides[49]. This segmentation process is based on the change in angular rotation of the shank of the leg, emulating heel strike during walking and running as well as segmenting any cyclic activity, including biking. The sagittal plane angular velocity of the IMU worn on the shank of the left leg was filtered with a fourth-order 6 Hz low-pass filter before detecting peaks. The largest relative peaks were required to be at least 0.5 s apart to identify one gait cycle.

The Apple Smartwatch data were stored on their paired smartphone and exported from the Health app. The data were split into files containing active energy burned, basal energy burned, and a variety of other information. Each estimate of energy expenditure contained a start and stop timestamp along with a value for kilocalories expended during that period. The energy expenditure rate in Watts was computed by dividing the kilocalories by the duration of time in seconds, and then multiplying by a scaling factor to convert the units to Watts. The active and basal energy estimates were interpolated and then summed to produce total energy expenditure estimates at 1 s intervals.

Data collection and processing used custom code relying on Python (version 3.6.1), Matlab 2019a, Motion Analysis 7.1, and OpenSim 4.0. Additional required python packages include numpy (1.17.4), scikit-learn (0.21.3), scipy (1.3.2), matplotlib (2.0.2), natsort (6.2.0), jupyter (1.0.0), ipython (5.3.0), and pandas (0.25.3). See the Code availability section for access to the public repository containing code and data to replicate the study.

**Estimation methods**. The methods of estimating energy expenditure we compared included: Steady-State Respirometry, Per-Breath Respirometry, Fast-Estimated Respirometry, the Activity Monitor, the Activity-Specific Model, the Data-Driven Model, the Musculoskeletal Model, the Smartwatch, and the Activity-Specific Smartwatch. These methods are capable of estimating energy expenditure for extended periods if the activity is within the aerobic threshold[15]. Directly measuring instantaneous energy expenditure of the whole body is not possible. Respirometry cannot measure instantaneous energy expenditure because of significant delays in replenishing energy stored in the muscles, primarily owing to mitochondrial dynamics, oxygen consumption, and blood circulation[50]. Activities outside the aerobic threshold violate steady-state assumptions in these respirometry methods and thus cannot be monitored accurately.

Steady-State Respirometry acted as the ground truth energy expenditure values for the steady-state conditions by averaging the last several minutes of respirometry data converted to energy expenditure with the Brockway equation. Conditions collected in the first three experiments lasted 5 mins and the steady-state estimate was averaged over the last 2 mins. Conditions collected in the last experiment to validate the Wearable System lasted 6 mins and the steady-state estimate was averaged over the last 3 mins.

Per-Breath Respirometry provided noisier estimates of energy expenditure once per breath also using the Brockway equation. Fast-Estimated Respirometry estimated steady-state quickly by fitting estimates per breath to a first-order exponential function. The asymptote value was treated as the estimate. Time-varying conditions typically do not have a known step change in energy expenditure, limiting use to steady-state conditions.

The Heart Rate Model estimated energy expenditure by using a linear regression model to estimate Steady-State Respirometry from steady-state heart rate values in beats per minute. The linear regression model had one weight and one bias parameter.

The Activity Monitor used a linear model to estimate energy expenditure based on the number of acceleration measurements that reached a threshold every minute. The threshold and count protocol followed the popular ActiGraph procedure. The accelerometer measured three axes and used the processing steps of the triaxial GT3X activity monitor. A previous regression equation was selected to scale the number of counts to energy expenditure in metabolic equivalents, which had units of kilocalories per kilogram per hour and were converted to Watts for comparison[37].

The Data-Driven Model used wearable sensor data to estimate energy expenditure once per stride. The model was fit using linear regression with ridge regularization with default regularization value of 1. Each stride of data were discretized to a fixed input size by splitting input signals into 30 bins, selected from previous experimentation[35]. The discretized input signals were flattened into a vector with single features including the subject's height, weight, and duration of each stride in seconds. The last 50 strides from each training condition were included in the training data. The training data was standardized separately across each input signal by subtracting the mean and dividing by the standard deviation. Tested conditions were normalized by using the mean and standard deviation from the training data. Interaction and nonlinear terms were not included due to the large number of input signals.

The Activity-Specific Model relied on ideal activity classification to estimate energy expenditure with separate models for each activity. Inputs to the models

consisted of subjects' height, weight, and stride duration. The models were linear regression with ridge regularization, using the default regularization parameter of 1. Although this method was compared with the Wearable System it was evaluated offline to add ideal activity classification.

The Musculoskeletal Model estimated energy expenditure by using a simulated anatomical model tracking measured kinematics and applying measured muscle activity to compute the metabolic rate of each leg muscle (Supplementary Fig. 2b). Owing to large computation times, estimates were computed from five strides per condition. The anatomical model was scaled by the subject's height and weight. The measured electromyography data were scaled to muscle activations using activation values from similar running conditions. Scaling to muscle activations during walking resulted in impossible activations above 100% during running[20,51]. Muscle parameters were computed using muscle-driven forward simulation relying on OpenSim software[19,52]. The normalized fiber length and velocity parameters were validated by comparing to related simulation work. The metabolic rate of each muscle was computed by passing the muscle activations and simulated muscle parameters to a metabolic probe model[22]. This simulation was determined to meet best practices[52,53] as the simulated net joint moments for the ankle and knee in the sagittal plane were within two standard deviations of experimental data during walking.

The Smartwatch estimated energy expenditure approximately every minute using an undisclosed model provided by Apple. The Smartwatch was an Apple watch series 1 (42 mm) with model number A1154 and software version 4.3.2. We evaluated the Apple watch series 1 because it is the only model that has been previously validated in other research experiments[31,39]. Apple does not provide information whether their method for estimating energy expenditure is different for the generations of Smartwatches. However, all versions of their Smartwatches have the same set of sensors and utilize the same software application, which we updated to the most recent available version. The estimates were exported from the Health app on a paired iPhone 6 S with model number A1633 and operating system version 13.2.3. The inputs to the model are unknown but may include wrist kinematics and heart rate. The Smartwatch was worn on the right wrist and calibrated following Apple's guidelines. The calibration process for each subject consisted of resetting the calibration, entering subject-specific information in the Health app, and ensuring a snug fit around the wrist. After all data were collected, the estimates were exported from the Apple Health app on the paired smartphone.

The Activity-Specific Smartwatch estimated energy expenditure approximately every 3 seconds using undisclosed and Activity-Specific Models provided by Apple. The Activity-Specific Smartwatch was an Apple watch series 1 (42 mm) with model number A1154 and software version 4.3.2. The estimates were exported from the Health app on a paired iPhone 6 with model number A1549 and operating system version 12.4.4. The Activity-Specific Smartwatch was worn on the left wrist. Before the start of each condition the user selected the appropriate model for that activity. The models for the walking, running, stair climbing, and biking activities were labeled as indoor walk, indoor run, stair stepper, and indoor cycling. The indoor run mode was selected for time-varying activities that included both walking and running.

**Performance metrics**. Absolute error measured the error between estimated and ground truth energy expenditure. The absolute error was calculated by finding the percent error between the estimate and ground truth, taking the absolute value of these errors, and then averaging all errors in each condition.

The relative error metric evaluated model precision in capturing the relative changes in energy expenditure across all conditions. The relative error first removed the difference between estimates and ground truth energy expenditure across all conditions for each subject, eliminating any consistent offset. The relative error was then calculated following the same steps as the absolute error.

The cumulative energy expenditure was calculated as the average energy expenditure over the last 3 mins of the 6-min condition, including any expenditure above quiet standing for 3 mins following the condition[44]. The cumulative energy expenditure approximated the total energy spent during the activity because methods relying on respirometry or heart rate had delayed response in the energy expenditure estimates and remained elevated for several minutes after a condition. The error of cumulative energy expenditure was computed between the estimates and ground truth following the same steps as the absolute error.

**Statistical analysis**. The method used to estimate energy expenditure had a significant effect for all reported error metrics with a Kruskal–Wallis one-way analysis of variance (fixed effect: method; $p \leq 3 \times 10^{-7}$). The Kruskal–Wallis test was selected because the errors did not follow a normal distribution. We then used paired $t$ tests to evaluate the significance of the reported errors. We compared the $t$ test results using a significance level of 0.05 and applied the Bonferroni correction with 15 comparisons, giving a modified significance level of 0.0033. The largest reported value was $p = 7 \times 10^{-4}$, indicating all results were statistically significant. There was no significant difference (paired $t$ test; $p = 0.64$) in the steady-state absolute error based on gender. Linear regression models that used the subject's heights, weights, or body mass index to estimate steady-state absolute error resulted in R-squared values of 0.009, 0.024, and 0.017, respectively. Thus, there was no significant correlation between the error and the subject's height, weight, or body mass index.

**Reporting summary**. Further information on research design is available in the Nature Research Reporting Summary linked to this article.

## Data availability
All materials necessary to replicate the results from this work are available in a public repository: https://simtk.org/projects/energy-est. This includes experimental data, including the series of experiments to evaluate different types of models, select the sensors for the Wearable System, collect training data for the Wearable System, and validate the Wearable System with a diverse population. Source data are provided with this paper. A reporting summary for this article is available as a Supplementary Information file.

## Code availability
All custom code necessary to replicate the results from this work is available in a public repository (https://simtk.org/projects/energy-est) and a permanent reference version on Github[54]. This includes code to estimate energy expenditure with the different models, perform musculoskeletal simulations, train the Data-Driven Model used by the Wearable System, validate the Wearable System results, and replicate the Wearable System.

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

## Acknowledgements

We thank S. Uhlrich, R. Jackson, and C. Welker for technical assistance and for maintaining the equipment used for this study. This work was supported by the National Science Foundation Graduate Research Fellowship Program Grant DGE-1656518, National Institutes of Health Grant U54EB020405, National Center for Simulation in Rehabilitation Research Grant P2C HD065690, and the Stanford Graduate Fellowship.

## Author contributions

P.S., M.K., S.D., and S.C. were responsible for study design and manuscript preparation. P.S. was responsible for data collection and analysis.

## Competing interests

The authors declare no competing interests.
