## [Peer Review File · Nature Communications]

Reviewer #1 (Remarks to the Author):

Review of "Sensing leg movement enhances wearable monitoring of energy expenditure"

The author suggested a wearable system to estimate energy expenditure during physical activities with lowering error by selecting data-driving model. The comparison between each model was will discovered and the fabricated wearable system seems to be will operated. Also, low cost of the wearable system is huge advantage for commercialization. However, the main content of this article is about data analysis, error minimization and system modeling, so it seems far from the scope pursued by Nature Communication. For this reason, I would recommend "REJECT" of this work.

Reviewer #2 (Remarks to the Author):

This paper presents an alternate method for estimating metabolic cost based on wearable sensors alone. Machine learning techniques were used to extract metabolic information from kinematics as measured by IMUs. This estimate was compared to several currently available estimate methods for both steady-state and speed-varying tasks. Analyses were performed for selection of sensors, models, and comparison to current techniques.

Major Revisions:

1. A unique aspect of this study is the attempt to estimate metabolic cost during time-varying activities. To make this argument, supporting literature and further rationale must be presented on how this has been done in the past and why it is reasonable that assuming steady-state values and interpolating between those values is a good estimate of metabolic cost for these activities.
2. Based on my understanding (paragraph in line 82), one main thrust of your argument is that data-driven wearable estimates have worked in the past; however, your method is new and innovative because you no longer require subject-specific information. If this is indeed the argument, two things need to be supported. First, the discussion needs to make it clear why your method did not require subject-specific information; in other words, how this new approach is less sensitive to anthropological differences. Second, whatever mechanism is proposed as an explanation (e.g. using lower body kinematics) should be set up in introduction as a method that has not been used before. Make it clear in this paragraph that previous data-driven models have not used something similar to your method before.
3. The hypothesis should be split into two separate hypotheses, one focusing on the efficacy of the data driven method and one on the wearable implementation of that method, or the hypothesis should reflect what you concluded: that the wearable system based on a data-driven model would perform best. If you split this into two hypotheses, this will make it clear why you have two major results and how those prove different things. Currently, it is confusing because you present the first result as selecting the method, but that doesn't make sense since your hypothesis was that the data-driven method would work best. Thus, you were not trying to select a method but determine whether your method actually performed better than other methods. If you do split this, then include statistics for this first study to back up the hypothesis that the data-driven method performed best. Then your second focus was to expand this system to a wearable device for testing. This second hypothesis incorporates selecting sensors, training the system, and then testing with new patients.

Minor Revisions:

Abstract:

In line 20, clarify which methods (smartwatch, respirometry, or both) estimate time-varying activities poorly.

In line 30, tie this point back to physical activity so it is clear how this could make an impact on patients' lives.

Introduction:

In the paragraph beginning on line 36 (beginning of the introduction), an explicit connection needs to be made between monitoring and participating. How does being able to monitor energy expenditure have a correlation to people's participation in physical activity?

In the sentence starting on line 108 (end of the introduction) where you argue your rationale for using lower body kinematics, the current argument for lower body vs. upper body is weak. Use the references from your limitation section (from the discussion) to state that upper body motion, "comprise[s] only a small portion of active energy expenditure".

Results:

Figure 2 makes the results seem like the substantial difference in accuracy is mostly in the resting phase post activity. This undercuts the main argument. It is useful to have the resting phase to demonstrate the latency in certain estimate methods; however, a close-up view of the dynamic condition demonstrating the superiority of the wearable estimate during steady-state would be beneficial.

When discussing Fast-Estimated Respirometry (line 152, 228) make it clear that this is not compared directly because it is not a wearable estimate. The purpose is instead to highlight that the wearable estimate is better at the beginning than even the fastest steady state laboratory estimator.

Discussion:

Address the following questions briefly: Why was only a linear regression model tried? Have other machine learning models been attempted? If not, why not?

The sentence beginning in line 329 is unclear.

The sentence in line 344 seems to be opposite the point of the paper. Even the most accurate activity recognition didn't work so why would one want to create an online version.

Materials and Methods:

The sentence in line 407 is poorly worded.

Reviewer #3 (Remarks to the Author):

In this paper, the authors face the problem of designing tools to assess the energy expenditure in common activities using wearable systems. The supported thesis is that data-driven methods, which are meant to rely on wearable measurements of lower limb kinematics and segmentation of input data by stride, would estimate more accurately the energy expenditure than current available

systems.

The general problem of assessing energy expenditure with wearables is widely discussed in literature. This paper has the great merit of taking into consideration, at the same time, several factors such as: four different input sources, several locations of input sources, and the methods exploited to extract energy expenditure.

Despite the limited sample (which is accounted for by the authors in the discussion section), the results support the hypothesis of the study. Moreover, the methodology used is interesting as per-se, since it takes into consideration at the same time, and in a very elegant way, the different factors affecting the accuracy of these systems.

My only suggestion regards table S.1. To facilitate the reading, it could be useful to add an initial column that points out the Number of sensors used: i.e. 4, 1, 2, 3. For each group, it could be interesting to comment how the lowest relative error is always obtained when the IMU input source is considered.

Finally, for further studies, I would suggest to include the respiration frequency in the analysis, see for example: <https://doi.org/10.3389/fphys.2017.00922> for the physiological implications of this variable and this: DOI: 10.1109/JSEN.2019.2899658 as an example of a wearable technology.

Regards,

Fabrizio Taffoni

Reviewer #4 (Remarks to the Author):

This manuscript describes the authors' endeavor at evaluating a wearable system based on IMUs for measuring energy expenditure. The authors compare their wearable system against a smart watch and conclude that body-worn IMUs fare better than a single-point smartwatch.

Overall, while the evaluation of body-worn IMUs against a single point smartwatch is useful for device engineers aiming to progress wearable health monitoring systems, it is not particularly revelatory that multiple sensors tightly strapped to the body provide higher accuracy than an integrated smart watch that can only measure metrics at one arbitrary point on the body. Furthermore, since the authors' use known, existing sensors in their wearable system, I don't personally see any added advances in device engineering that recommend this work. This work seems suited for a more specialized journal rather than a broad-audience journal

Some specific comments:

1. P3-L130: "We measured the ground truth energy expenditure by averaging the respirometry measurements from the last three minutes of each condition, which we refer to as Steady-State Respirometry."

(a) What is the reason behind choosing respirometry measurements as the ground truth energy expenditure? Is it because of the lower relative error (%) compared to other methods (fig S3A)? Why not other measurement techniques? It requires more clarification.

(b) Based on fig S3B, the respirometry error plot does not reveal any steady-state behavior even after 74 seconds. It is suggested to prove this behavior for the last 3 minutes of each condition if it is claimed so.

(c) Why did the authors not choose a ground truth that enables direct measurements for transitions between walking and running rather than requiring interpolation?

2. L173- "An IMU on the shank and thigh of one leg were found to have the lowest steady-state error"

Please back up this statement with data or a reference.

3. L267- "This wearable system is designed to estimate energy expenditure in real-time for everyday use."

This system is, in actuality, a number of IMUs that are strapped tightly to the lower body, similar to other commercial heart rate monitoring iterations and is not particularly transformative or imaginative in implementation. This system will not be widely adopted to daily life unless it provides users with comfortability and ease of application. The readers must be provided with some details/data on how comfortable and unobtrusive the subjects find this wearable system during their physical activities. I understand that "comfort" and "unobtrusive-ness" are hard, non-standardized metrics to measure, but there are ways to probe the useability of the authors' system, for example user studies with at least 20 people.

4. L 162 & L198 & L216- Previously in fig S6 the authors reported that the Data-Driven model resulted in 73% error in estimating running conditions. What if this model results in the same or even higher amount of error (%) for estimating energy expenditure in biking condition? How do the authors validate the results in both cases?

5. The authors have not mentioned the effect of duration of the activity on the energy expenditure, and how accurate different models can monitor/estimate it.

RESPONSE TO REVIEWERS

Reviewer #1:

The author suggested a wearable system to estimate energy expenditure during physical activities with lowering error by selecting data-driving model. The comparison between each model was will discovered and the fabricated wearable system seems to be will operated. Also, low cost of the wearable system is huge advantage for commercialization. However, the main content of this article is about data analysis, error minimization and system modeling, so it seems far from the scope pursued by *Nature Communication*. For this reason, I would recommend “REJECT” of this work.

We thank the reviewer for their time in reviewing this work, and appreciate their recognition of this research as thorough and with a well designed system that may be easily extended to commercial use. We believe that this study makes contributions that will be of interest to a broader audience, including members of the fields of biomechanics, machine learning, electromechanical systems engineering, and clinical treatment of obesity and gait disorders, which will be of interest to the diverse readership of *Nature Communications*. The main contribution is an inexpensive and accurate approach to estimate energy expenditure, which we expect to have broad appeal because it could be used to overcome challenges in accurately monitoring physical activity at large scale, studying relationships between activity and health, and managing weight. The system is open-source and can be replicated easily, facilitating application to products that address these challenges. This work also makes a scientific contribution through rigorous experiments to select the sensors that relate most closely to energy expenditure, which will be of great interest to biomechanics and physiology researchers. The method of device design through sensor selection also makes a contribution, and could be applied to other types of wearable devices. We hope that the reviewer will consider this revised manuscript, in which we have tried to better explain these aspects of the significance of the work.

Reviewer #2:

This paper presents an alternate method for estimating metabolic cost based on wearable sensors alone. Machine learning techniques were used to extract metabolic information from kinematics as measured by IMUs. This estimate was compared to several currently available estimate methods for both steady-state and speed-varying tasks. Analyses were performed for selection of sensors, models, and comparison to current techniques.

Thank you for your thoughtful review of this manuscript. We have revised the manuscript substantially in response to your insightful comments.

Major Revisions:

1. A unique aspect of this study is the attempt to estimate metabolic cost during time-varying activities. To make this argument, supporting literature and further rationale must be presented on how this has been done in the past and why it is reasonable that assuming steady-state values and interpolating between those values is a good estimate of metabolic cost for these activities.

We agree that estimating time-varying activities is a unique and challenging application. Ground truth estimates of energy expenditure during time-varying activities are challenging to obtain because respirometry measurements have a significant delay and slow sampling rate. We have added an additional reference and description to more clearly define the errors associated with estimating the cumulative and instantaneous energy expenditure during time varying motion starting in line 354:

“The time-varying conditions had a period of 30 seconds, faster than the delays in breath-to-breath measurements. The 30 second period was selected to minimize energy expenditure associated with changing speeds by maintaining a low average acceleration of 0.07 m/s^2 . Prior experiments have found that varying walking speeds by 0.6 m/s sinusoidally with a 4 second period (0.15 m/s^2 peak acceleration) increases average energy expenditure by 4% to 8% (44). Assuming that increased energy cost is linearly related to acceleration, the cumulative energy expenditure during time-varying conditions would be about 2% to 4% higher than the interpolated estimates of instantaneous energy expenditure, because they interpolate between steady-state energy expenditure which do not include the costs of additional acceleration.”

(44). N. Seethapathi, M. Srinivasan. The metabolic cost of changing walking speeds is significant, implies lower optimal speeds for shorter distances, and increases daily energy estimates. *Biology letters* **11**, 9 (2015).

2. Based on my understanding (paragraph in line 82), one main thrust of your argument is that data-driven wearable estimates have worked in the past; however, your method is new and innovative because you no longer require subject-specific information.

Our intention for the specified paragraph (now starting in line 84) was that we do not require subject-specific training data. Here, subject-specific training data indicates that a subject has to complete a series of experiments in a laboratory with respirometry so that a model can be customized for that subject. Many of the existing data-driven wearable methods require this subject-specific training data and evaluate their model's accuracy on these same subjects. This results in models that are overfit to the data from these specific subjects and will not perform as well on a general population, thus their reported accuracy is misleading. We have taken a more general approach and have clarified this in the sentences starting in lines 84 and 91.

84: Combining wearable sensors and data-driven methods enables portable and computationally efficient estimation, but many methods rely on subject-specific data to train their models and do not evaluate the accuracy for new subjects.

91: Wearable data-driven methods using subject-specific training data have estimated energy expenditure with relatively low errors of 14% to 27% (16, 28, 30). Unfortunately, methods using subject-specific training data have about twice the expected error when estimating energy expenditure for new subjects (35).

Our system uses the subject's height and weight, which many other methods also include. However, our system does not use subject-specific training data.

If this is indeed the argument, two things need to be supported. First, the discussion needs to make it clear why your method did not require subject-specific information; in other words, how this new approach is less sensitive to anthropological differences. Second, whatever mechanism is proposed as an explanation (e.g. using lower body kinematics) should be set up in introduction as a method that has not been used before. Make it clear in this paragraph that previous data-driven models have not used something similar to your method before.

In the revised manuscript, we have clarified that our model uses a few pieces of easily-obtained subject-specific information (height and weight). We have also edited the text to more clearly convey how this method is unique and achieves significantly lower errors. We believe the difference has two main causes: 1) the careful selection of wearable sensors on the lower limbs that capture information more directly related to energy expenditure and 2) the estimation of energy expenditure once per gait cycle to create time-invariant and rapid estimates that help relate the changes in lower-limb motion during each step to changes in energy expenditure. We have made these two key differences clearer in the last paragraph of the introduction in the sentences starting in lines 110 and 116.

110: Lower-limb kinematics could provide more information than heart rate, wrist kinematics, or trunk kinematics because lower-limb activities are more common, have large energy expenditure, and have significant leg motion compared to upper-limb activity (7, 8).

116: This new modeling approach creates time-invariant inputs appropriate for simple data-driven models such as linear regression which requires minimal computation time.

3. The hypothesis should be split into two separate hypotheses, one focusing on the efficacy of the data driven method and one on the wearable implementation of that method, or the hypothesis should reflect what you concluded: that the wearable system based on a data-driven model would perform best. If you split this into two hypotheses, this will make it clear why you have two major results and how those prove different things.

Currently, it is confusing because you present the first result as selecting the method, but that doesn't make sense since your hypothesis was that the data-driven method would work best. Thus, you were not trying to select a method but determine whether your method actually performed better than other methods. If you do split this, then include statistics for this first study to back up the hypothesis that the data-driven method performed best. Then your second focus was to expand this system to a wearable device for testing. This second hypothesis incorporates selecting sensors, training the system, and then testing with new patients.

We agree with your second recommendation that the hypothesis should reflect what we concluded: that the Wearable System based on a data-driven model would perform best. Our goal was to test the hypothesis that the data-driven model performed better than the other methods of estimation, rather than perform a test to specifically select the data-driven model from among the compared models. We have rephrased the language in all sentences related to the "selecting the wearable energy expenditure method" to become "evaluating the wearable data-driven method", to focus on this comparison between the data-driven model and other comparable methods and remove it from the original hypothesis. We have rephrased the sentences starting in the following lines. We feel this more closely reflects the original hypothesis described in the abstract.

- 127: Evaluating the wearable data-driven method
- 150: We validated that the Data-Driven Model using all wearable sensor inputs was the best wearable method for estimating energy expenditure because it had the lowest errors during steady-state and time-varying conditions.
- 443: We hypothesized that the Wearable System would have significantly lower error than the state-of-the-art methods.
- 458: Evaluating the wearable data-driven method and selecting informative sensors
- 459: In order to evaluate the Data-driven method and select the fewest sensors necessary to estimate energy expenditure, we performed an experiment to collect data from a variety of conditions and wearable sensors for offline testing.
- 480: The wearable data were used to compare a variety of methods including: ...
- 595: The methods of estimating energy expenditure we compared included: ...

Minor Revisions:

Abstract:

In line 20, clarify which methods (smartwatch, respirometry, or both) estimate time-varying activities poorly.

We have changed line 20 to clarify that wearable methods (including smartwatches and others mentioned in the introduction) estimate the energy expenditure during time-varying activities poorly. Here we refer to error in cumulative energy expenditure, which uses respirometry as an accurate ground truth. All existing methods (lab-based and wearable) perform poorly for estimating instantaneous energy expenditure during time-varying activities, but there is no ideal benchmark for directly measuring this instantaneous change.

20: Existing wearable methods estimate time-varying activities poorly, which comprise 40% of daily steps.

In line 30, tie this point back to physical activity so it is clear how this could make an impact on patients' lives.

We agree that the tie back to physical activity should be clearer. We have changed line 30 to "This approach enables accurate physical activity monitoring, which could enable new energy balance systems for weight management or tools for large-scale activity monitoring."

Introduction:

In the paragraph beginning on line 36 (beginning of the introduction), an explicit connection needs to be made between monitoring and participating. How does being able to monitor energy expenditure have a correlation to people's participation in physical activity?

Our goal in the first sentence of the introduction was to inform the reader that the high mortality rate due to physical inactivity was the motivation for wanting to understand how physical activity relates to health by using methods for monitoring. We have clarified this in line 37: "Effective physical activity monitoring is necessary to understand and overcome inactivity, which is the fourth largest cause of mortality".

In the sentence starting on line 108 (end of the introduction) where you argue your rationale for using lower body kinematics, the current argument for lower body vs. upper body is weak. Use the references from your limitation section (from the discussion) to state that upper body motion, "comprise[s] only a small portion of active energy expenditure".

We agree that the motivation for relying on lower-limb kinematics should make it clear that lower-limb activities have higher energy expenditure than upper-limb activities. We have updated the sentence starting on line 110 to address this and added supporting references: "Lower-limb kinematics could provide more information than heart rate, wrist kinematics, or trunk kinematics because lower-limb activities are more popular for exercise and have larger energy expenditure than upper-limb activities (7, 8)."

Results:

Figure 2 makes the results seem like the substantial difference in accuracy is mostly in the resting phase post activity. This undercuts the main argument. It is useful to have the resting phase to demonstrate the latency in certain estimate methods; however, a close-up view of the dynamic condition demonstrating the superiority of the wearable estimate during steady-state would be beneficial.

We agree that Figure 2 shows substantial differences between the wearable methods and ground truth energy expenditure during the standing rest. Figure 2 also represents large differences between the energy expenditure estimates of the wearable methods and ground truth throughout the steady-state conditions. To better highlight these steady-state differences we have changed the description of Figure 2B in line 900: “The error as a function of time from the start of the condition evaluated how quickly the methods reached a steady-state estimate, averaged across all conditions”. Thus, Figure 2B represents the average steady-state error as a function of time from the start of the condition. This highlights the relative performance difference between the Wearable System and the other wearable methods. The discussion in line 343 has been updated to reflect this: “The Wearable System estimates were accurate from the start of steady-state conditions (Fig. 2B)”.

We have also improved the discussion of the steady-state differences between the wearable methods to more clearly identify which activities illustrate large steady-state errors for each wearable method in Figure 2A and Figure S7, which breaks down the steady-state error for each method and each condition. These changes start in line 307 of the discussion: “The Activity-Specific Smartwatch exhibited similar trends to the Heart Rate Model, including higher estimates during walking and quiet standing, estimates drifting during conditions, and a delayed response to changes in energy expenditure (Fig. 2A). The Smartwatch estimates during stair climbing and biking were substantially lower than ground truth values, possibly because subjects held onto safety railings which minimized wrist motion (Fig. S7)”.

When discussing Fast-Estimated Respirometry (line 152, 228) make it clear that this is not compared directly because it is not a wearable estimate. The purpose is instead to highlight that the wearable estimate is better at the beginning than even the fastest steady state laboratory estimator.

We agree that Fast-Estimated Respirometry should clearly be noted as a laboratory based estimate. We have updated line 155 (“Fast-Estimated Respirometry had the lowest steady-state error after 74 seconds, confirming its usefulness as a laboratory-based test where accuracy is paramount and longer trial times are acceptable”) and 230 (“The Wearable System had the lowest error for the first 44 seconds, after which the laboratory-based, Fast-Estimated Respirometry had the lowest error”) to reflect this.

Discussion:

Address the following questions briefly: Why was only a linear regression model tried? Have other machine learning models been attempted? If not, why not?

During our experiment to select the best method for estimating energy expenditure we prototyped other machine learning models such as Neural Networks, but found that these models too easily overfit the training data and did not provide any improvement over linear regression. Future studies that have access to a larger amount of training data may find them useful. In the limitations on line 409 we have added this discussion: “Early evaluations of other machine learning models performed worse than linear regression, likely because they overfit to the relatively small amount of training data. Future studies with access to more data may benefit from a more expressive model, such as a Neural Network.”

The sentence beginning in line 329 is unclear.

We have changed this sentence into several sentences, in order to clarify the experimental details of how the time-varying conditions were collected. Beginning in line 354: “The time-varying conditions had a period of 30 seconds, faster than the delays in breath-to-breath measurements. The 30 second period was selected to minimize energy expenditure associated with changing speeds with a low average acceleration of 0.07 m/s^2 .”

The sentence in line 344 seems to be opposite the point of the paper. Even the most accurate activity recognition didn't work so why would one want to create an online version.

We agree that the suggestion that the activity classification method could be used online is not relevant to the purpose of the text, which should focus on its poor performance. This sentence was removed and the detail that the ideal activity classification was “performed manually” was added to line 373:

“The Wearable System had lower steady-state error than the Activity-Specific Model, however, which was based on ideal activity classification performed manually, indicating the Wearable System may extract additional information such as how the activity was performed, where changes in motion may relate to muscle-level energy expenditure.”

Materials and Methods:

The sentence in line 407 is poorly worded.

We agree this was confusing and have revised this sentence and shifted this text down one sentence starting in line 455 as “These sections describe the motivation for the experiment, which sensors were used to collect data, and how the data were collected”.

Reviewer #3:

In this paper, the authors face the problem of designing tools to assess the energy expenditure in common activities using wearable systems. The supported thesis is that data-driven methods, which are meant to rely on wearable measurements of lower limb kinematics and segmentation of input data by stride, would estimate more accurately the energy expenditure than current available systems. The general problem of assessing energy expenditure with wearables is widely discussed in literature. This paper has the great merit of taking into consideration, at the same time, several factors such as: four different input sources, several locations of input sources, and the methods exploited to extract energy expenditure. Despite the limited sample (which is accounted for by the authors in the discussion section), the results support the hypothesis of the study. Moreover, the methodology used is interesting as per-se, since it takes into consideration at the same time, and in a very elegant way, the different factors affecting the accuracy of these systems.

Thank you for your thoughtful review of our manuscript.

My only suggestion regards table S.1. To facilitate the reading, it could be useful to add an initial column that points out the Number of sensors used: i.e. 4, 1, 2, 3. For each group, it could be interesting to comment how the lowest relative error is always obtained when the IMU input source is considered.

We agree and have added this number next to the name of the sensor class at the top of each column and included a description in the caption of table S1 (Supplementary Materials, line 84): “The number next to the sensor classes indicates the number of sensors used for that class.” We also agree that it is helpful to point out that including the IMUs with any other sensor classes reduced the error. We have added that to the caption for table S1 as well (Supplementary Materials, line 85): “The individual class with the lowest error was IMU and in general the IMUs paired with other sensors had lower error than the other sensors alone.”

Finally, for further studies, I would suggest to include the respiration frequency in the analysis, see for example: <https://doi.org/10.3389/fphys.2017.00922> for the physiological implications of this variable and this: DOI: 10.1109/JSEN.2019.2899658 as an example of a wearable technology.

Thank you for this suggestion. In the discussion paragraph on future work we now address respiration frequency as a candidate measure for examination in future studies and refer to the recommended paper starting line 419: “Evaluating other possible signals, such as respiration frequency (47), may help differentiate energy expenditure for conditions with similar kinematics.”

47. A. Nicolò, C. Massaroni, L. Passfield. Respiratory frequency during exercise: the neglected physiological measure. *Frontiers in physiology* **8**, 922 (2017).

Reviewer #4 (Remarks to the Author):

This manuscript describes the authors' endeavor at evaluating a wearable system based on IMUs for measuring energy expenditure. The authors compare their wearable system against a smart watch and conclude that body-worn IMUs fare better than a single-point smartwatch.

Overall, while the evaluation of body-worn IMUs against a single point smartwatch is useful for device engineers aiming to progress wearable health monitoring systems, it is not particularly revelatory that multiple sensors tightly strapped to the body provide higher accuracy than an integrated smart watch that can only measure metrics at one arbitrary point on the body. Furthermore, since the authors' use known, existing sensors in their wearable system, I don't personally see any added advances in device engineering that recommend this work. This work seems suited for a more specialized journal rather than a broad-audience journal.

We agree that, from an information theoretic perspective, multiple sensors would be expected to provide data sufficient for at least as accurate estimation of energy expenditure as a single sensor. Similarly, we agree that the freedom to choose sensor location should be expected to result in at least as accurate an estimate as a sensor with a prescribed location. We now address these ideas in the Discussion starting in line 317 (and quoted in context below).

However, without this study, we could not have known how much of an improvement was possible, nor which signals are important to measure, nor which body locations were best for measuring them.

For example, it might have been that alternate sensor configurations yielded only small improvements in accuracy, as found in prior studies using many sensors with subject-specific calibration. Instead we found that just two well-placed sensors, using a well-designed data-driven model and no subject-specific training data, could be used to estimate energy expenditure with about three times lower error than smartwatch methods, which are used by millions of individuals. Using the approach described in this manuscript with inputs from just one of these two well-placed sensors still resulted in less than half the error of the smartwatches. We think that this is a surprisingly large improvement in performance with important practical implications for both experimental tools and future products for monitoring energy expenditure.

As another example, it might have been that physiological signals, such as heart rate, were most effective for estimating energy cost. In that case, the wrist would have proven an excellent location for data collection, given the relative ease of measuring pulse. Instead we found that measurement of signals related to thigh and shank movement provided the best information for estimating energy expenditure, and that inclusion of heart rate did not improve the accuracy of the estimate. This is surprising, given that most prior approaches have incorporated heart rate.

As a final example, placing sensors at many body locations might have been required to accurately estimate energy expenditure. The use of musculoskeletal models to estimate

instantaneous muscle energy expenditure would require relatively accurate estimates of joint kinematics and muscle activity, derived from inertial measurement units on each body segment and electrodes on the belly of each major muscle group. Instead we found that only two inertial measurement units on the thigh and shank provided nearly equal accuracy to many sensors of many types distributed across the body. This is also a surprising result with high utility for device design.

We now address these ideas more clearly in the Discussion starting in line 317:

“The finding that the Wearable System provides significantly more accurate energy expenditure estimates than existing wearable methods by using two carefully selected sensors is surprising. From an information theoretic perspective, we would expect that selecting multiple sensors would achieve at least the same accuracy of a single prescribed sensor, such as a smartwatch. However, the careful selection of two IMU sensors enabled the Wearable System to have three times lower cumulative error than the smartwatches. A version with inputs from the single thigh IMU still had less than half the cumulative error of the smartwatches. Including heart rate, a signal used in many wearable methods for estimating energy expenditure, as an input to the Wearable System did not improve the accuracy. Even using comprehensive sensor measurements of leg kinematics and major muscle groups performed similarly to the two selected IMU sensors. When designing wearable devices, rigorous sensor selection may provide counterintuitive results that can significantly improve performance.”

We also agree that development of sensor hardware is not an important contribution of this study. Instead, selection of sensor type and location, and formulation of an appropriate algorithm for interpreting sensor data, make an important contribution both to the scientific understanding of indicators of energy expenditure and to the engineering understanding of the design requirements of wearable devices that can accurately estimate energy expenditure. We have clarified this intended contribution at the end of the Introduction in line 119:

“To evaluate this hypothesis, we performed experiments to select two sensors worn on one leg from a comprehensive set of existing wearable biomechanics sensors.”

We believe this work is of interest to a broad audience because it makes contributions across several fields. Clinical researchers studying obesity and physical activity may be interested in this new method for more accurately estimating metabolic energy use. Engineers at companies serving this clinical community may find an opportunity for a new product using this method. Researchers studying machine learning may be interested in this new application area, as well as insights into why this form of data segmentation proved more effective than prior approaches. Biomechanics researchers may be interested in the implications of the selected sensor signals and locations for monitoring and classifying gait, or for better understanding the mechanics underlying energy consumption in human locomotion. We have clarified the utility of this study to these groups in the Discussion starting in line 428:

“Clinical researchers studying obesity and physical activity could use systems like this one to more accurately investigate how activity relates to health outcomes and inform health policies. Engineers designing wearable devices could leverage our sensor selection approach to improve the effectiveness of their devices. Machine learning researchers may find our insights into using gait cycle structure to create time-variant inputs helpful for modeling other challenges related to motion and physiology. Biomechanics researchers may be interested in the implications of the selected sensor signals and locations for monitoring and understanding gait and energy expenditure.”

Some specific comments:

1. P3-L130: “We measured the ground truth energy expenditure by averaging the respirometry measurements from the last three minutes of each condition, which we refer to as Steady-State Respirometry.”

(a) What is the reason behind choosing respirometry measurements as the ground truth energy expenditure? Is it because of the lower relative error (%) compared to other methods (fig S3A)? Why not other measurement techniques? It requires more clarification.

We selected respirometry because it is the gold-standard in clinical experiments due to its consistency and low-error (1-3%) when measuring steady-state energy expenditure during activities below the aerobic threshold. We more clearly address why we selected respirometry as the ground truth in the introduction in line 73:

“Laboratory-based methods accurately estimate energy expenditure during steady-state activities but are not feasible for everyday use. Respirometry requires minutes of breath-based measurements from expensive and intrusive equipment for steady-state estimation (15, 16)”.

Fast-Estimated Respirometry discussed in figure S3A is an approximation of steady-state respirometry used to provide faster, but higher error, estimates of steady-state energy expenditure. We have clarified the difference between steady-state respirometry and Fast-Estimated Respirometry in the following lines:

- 133: We measured ground truth energy expenditure by averaging respirometry measurements from the last three minutes of each condition, which we refer to as Steady-State Respirometry.
- 140: We estimated metabolic energy expenditure with several methods including: the Heart Rate Model, the Activity Monitor, the Musculoskeletal Model using muscle-level energy estimates, the Data-Driven Model using all wearable sensor data segmented by stride in a linear regression model (fig. S2), Per-Breath Respirometry, and Fast-Estimated Respirometry which fit lab-based respirometry measurements to a first-order exponential function for quicker steady-state estimates.

- Figure S3A description in line 34: Fast-Estimated Respirometry relies on the lab-based respirometry equipment which provides ground truth energy expenditure measurements.

(b) Based on fig S3B, the respirometry error plot does not reveal any steady-state behavior even after 74 seconds. It is suggested to prove this behavior for the last 3 minutes of each condition if it is claimed so.

We agree, the respirometry measurements in figure S3B continue to fluctuate during the time-varying activity where the user changes walking and running speed. The period for this time-varying activity is 48 seconds and the rise time of respirometry is approximately 42 seconds (43). Thus, respirometry estimates will fluctuate during this activity. However, averaging the last 3 minutes of respirometry measurements will include approximately 4 periods of the time-varying profile, providing an accurate estimate of the cumulative energy expenditure. Averaging the last 3 minutes of respirometry measurements for time-varying walking is the best practice for measuring energy expenditure, since a steady-state value cannot be obtained (44). We have added a new supporting reference (44) and updated the Methods in line 687:

“The cumulative energy expenditure was calculated as the average energy expenditure over the last 3 minutes of the 6-minute condition, including any expenditure above quiet standing for 3 minutes following the condition (44).”

44. N. Seethapathi, M. Srinivasan. The metabolic cost of changing walking speeds is significant, implies lower optimal speeds for shorter distances, and increases daily energy estimates. *Biology letters* **11**, 9 (2015).

(c) Why did the authors not choose a ground truth that enables direct measurements for transitions between walking and running rather than requiring interpolation?

Respirometry measurements reflect energy expended by the body with significant delays due to biological processes. Thus, it is impossible to directly measure instantaneous energy expenditure during time-varying activity, such as when transitioning between walking and running. We have added an additional reference and clarified this in the Methods in line 599:

“Directly measuring instantaneous energy expenditure of the whole body is not possible. Respirometry cannot measure instantaneous energy expenditure because of significant delays in replenishing energy stored in the muscles, primarily due to mitochondrial dynamics, oxygen consumption, and blood circulation (50).”

(50). P. Krstrup, A. M. Jones, D. P. Wilkerson, J. A. Calbet, J. Bangsbo. Muscular and pulmonary O₂ uptake kinetics during moderate and high intensity submaximal knee extensor exercise in humans. *The Journal of physiology* **587**, 1843-1856 (2009).

To better understand the error in our method of interpolating instantaneous energy expenditure we have added the following discussion starting in line 354: “The 30 second period was selected to minimize energy expenditure associated with changing speeds with a low average acceleration of 0.07 m/s². Prior experiments found that varying walking speeds by 0.6 meter per second sinusoidally with a 4 second period (0.15 m/s² acceleration) increased the energy expenditure by 4% to 8% (44). Assuming the increased energy cost is linearly related to acceleration, the cumulative energy expenditure estimates should have a small error of 2% to 4%. By extension, the interpolated estimates of instantaneous energy expenditure may accurately reflect the change in energy expenditure as a function of walking speed, but slightly lower across all speeds due to the added cost of acceleration”.

2. L173- “An IMU on the shank and thigh of one leg were found to have the lowest steady-state error”. Please back up this statement with data or a reference.

We have included an additional supplementary table S2 that includes the steady-state errors of models for all permutations of the IMUs which was used to select the IMU on the thigh and shank. We have updated the language in line 179: “We then compared all permutations of the inertial measurement units, each of which had a triaxial accelerometer and gyroscope (table S2). The best results were achieved with one sensor on the shank and one on the thigh.”

Location of IMU Placements				
Hip	Thigh	Shank	Foot	Relative Error (%)
X	X	X	X	13.7
X				16.8
	X			16.7
		X		17.1
			X	17.4
X	X			14.7
X		X		16.0
X			X	17.9
	X	X		13.7
		X	X	15.7
	X		X	14.9
X	X		X	14.6
X		X	X	14.5
X	X	X		14.8
	X	X	X	15.4

Table S2. The Data-Driven Model results for permutations of inertial measurement units (IMUs) placed at different locations on one leg. The IMU placements included the hip, thigh, shank, and foot. The permutation with the lowest error was with one IMU worn on the thigh and shank. The error was computed using the cross-validation approach over all permutations of subjects and conditions.

3. L267- "This wearable system is designed to estimate energy expenditure in real-time for everyday use."

This system is, in actuality, a number of IMUs that are strapped tightly to the lower body, similar to other commercial heart rate monitoring iterations and is not particularly transformative or imaginative in implementation. This system will not be widely adopted to daily life unless it provides users with comfortability and ease of application. The readers must be provided with some details/data on how comfortable and unobtrusive the subjects find this wearable system during their physical activities. I understand that "comfort" and "unobtrusive-ness" are hard, non-standardized metrics to measure, but there are ways to probe the useability of the authors' system, for example user studies with at least 20 people.

We agree that this statement does not accurately reflect our evaluation of the Wearable System. We did not test the Wearable System in an at-home use case. Rather, we evaluated it in a lab setting within constraints that are consistent with the challenges of real-time, everyday use. The Wearable System was designed to be a research project and not a finished consumer product. We have changed line 278:

"The Wearable System used low-cost inertial measurement units worn on the shank and thigh of one leg to estimate energy expenditure in real-time. This approach is portable and unobtrusive, meeting the criteria to allow for everyday use. We demonstrated that the Wearable System accurately estimated energy expenditure during steady-state and time-varying activities including walking, running, stair climbing, and biking, the activities that contribute the most to daily energy expenditure."

We agree that understanding how a system may be adopted in real-life is important. To address this we have performed a post-hoc survey of 21 subjects that participated in the validation of the Wearable System. The questions consisted of a standardized system usability survey (40) (table S4) and system comfort survey (41) (table S5). The results of these tables have been added to the paragraph starting in line 265:

"Subject surveys found the Wearable System to be comfortable and have high usability. 21 subjects who participated in the validation of the Wearable System were surveyed. The usability was evaluated with the System Usability Scale (40). The Wearable System had a relatively high overall score of 80.9 out 100 averaged across subjects (table S4), which is the 90th percentile among 5,000 device surveys that used the System Usability Scale (41). The comfort related metrics were evaluated with a survey based on the Questionnaire for User Interaction Satisfaction (42). The Wearable System had high scores associated with different comfort related attributes (table S5). This indicates the Wearable System, which is a research prototype has potential for use in clinical or at-home settings".

Additional details have been added in the Methods section starting in line 552: "A post-hoc survey of 21 subjects was used to evaluate the usability of the Wearable System as well as metrics related to comfort. The usability questionnaire was the standard System Usability Scale

(40), which is a Likert scale meant to evaluate the usability of a system. 21 participants evaluated the Wearable System with this survey. The metrics related to comfort were based on the Questionnaire for User Interaction Satisfaction survey (42).”

- (40). J. Brooke. SUS: A quick and dirty usability scale. *Usability evaluation in industry* **189**, (1996).
- (41). J. Sauro. A practical guide to the System Usability Scale: Background, benchmarks, and best practices. *Measuring Usability LLC*, (2011).
- (42). J. P. Chin, V. A. Diehl, K. L. Norman. Development of an instrument measuring user satisfaction of the human-computer interface. *In Proceedings of the SIGCHI conference on Human factors in computing systems*, 213-218 (1988).

Question text (1 = Strongly Agree, 2 = Somewhat Agree, 3 = Neither Agree Nor Disagree, 4 = Somewhat Disagree, 5 = Strongly Disagree)	Mean ± Std
I think that I would like to use this system frequently.	2.8 ± 1.1
I found the system unnecessarily complex.	4.4 ± 0.8
I thought the system was easy to use.	1.6 ± 0.6
I think that I would need the support of a technical person to be able to use this system.	4.5 ± 0.5
I found the various functions in this system were well integrated.	1.6 ± 0.8
I thought there was too much inconsistency in this system.	4.4 ± 0.8
I would imagine that most people would learn to use this system very quickly.	1.6 ± 0.8
I found the system very cumbersome to use.	3.8 ± 1.1
I felt very confident using the system.	1.8 ± 0.9
I needed to learn a lot of things before I could get going with this system.	4.6 ± 0.6
Total usability score (out of 100)	80.9 ± 7.2

Table S4. System usability survey results. This questionnaire is the standard System Usability Scale (40). It is a Likert scale meant to evaluate the usability of a system. 21 participants evaluated the Wearable System with this survey. The Wearable System had a relatively high usability score of 80.9 averaged across participants.

Prompt: Overall reactions to the Wearable System	Mean ± Std
0 (heavy) to 9 (light)	7.3 ± 1.3
0 (obtrusive) to 9 (unobtrusive)	5.9 ± 2.3
0 (complex) to 9 (simple)	7.0 ± 1.9
0 (painful) to 9 (comfortable)	7.3 ± 1.1
0 (bulky) to 9 (compact)	6.6 ± 2.0

Table S5. System comfort survey results. This questionnaire was based on the Questionnaire for User Interaction Satisfaction survey (42). 21 participants were asked to evaluate the Wearable System on a scale of 0 to 9 for different comfort related metrics. The Wearable System had high values, indicating it was comfortable to wear and use.

The discussion of these tables has been added in the paragraph starting in line 390: “The approach used by the Wearable System may be effective for monitoring physical activity, but improved hardware would be necessary for large scale deployment. The Wearable System was a proof-of-concept device with a bulky microcontroller and wired sensors. Despite being a prototype, the subject surveys reported that the Wearable System had a high usability rating (table S4) and high ratings in metrics related to comfortability (table S5). This is likely due to the light weight and small form factor of the IMUs. A demonstration of the processing for donning and doffing the device shows this simple procedure, respectively taking 25 and 14 seconds (Movie S4). A future product would need to be refined by using common sensors such as a smartphone worn on the thigh or embedded in one shoe, integrating into clothing, or using stickers powered by heat or sweat (45). A smartphone could provide wireless sensor communication and compute estimates. This future product will also require investigation to determine if it is convenient enough for large scale, everyday use.”.

A supplementary video was added to show the Wearable System being donned and doffed in 25 and 15 seconds, respectively (movie S4). This highlights the simplicity for use in clinical or at-home settings and potential for incorporation into clothing or a commercial product.

4. L 162 & L198 & L216- Previously in fig S6 the authors reported that the Data-Driven model resulted in 73% error in estimating running conditions. What if this model results in the same or even higher amount of error (%) for estimating energy expenditure in biking condition? How do the authors validate the results in both cases

Our goal in figure S6 was to explore the performance of the Data-Driven model when estimating a new activity for which the model does not have any similar training data. This is challenging because the model relies on having training data similar to any activities it will be tested with. Running estimates were poor, likely because running had a significantly higher energy expenditure than the other walking conditions. We have clarified this point in line 164: “Holding out and evaluating running conditions resulted in the largest errors because the model estimated smaller energy expenditure values similar to the training data (fig. S6).”

To address your question of how well a Data-driven model could estimate biking without similar training data, we performed an additional test where we constructed a dataset consisting of one condition of walking, running, stair climbing, and biking. We iteratively held out each condition from the training data, trained a model, and evaluated that condition. The average errors for walking, running, stair climbing, and biking were 25.3%, 55.5%, 24.8%, and 17.2%. These results are similar to those reported in the main text, in which the running condition had the highest error, likely due to the larger energy expenditure value being outside the range of the training data, and thus difficult to estimate.

We validate the Wearable System’s errors associated with each condition in Figure S7. This shows the Wearable System performs similarly with relatively low error for all conditions, with the largest error being approximately 20%. Thus, the Wearable System has sufficient training

data for accurately estimating energy expenditure across the range of conditions tested in this study.

5. The authors have not mentioned the effect of duration of the activity on the energy expenditure, and how accurate different models can monitor/estimate it.

All methods described here can monitor activity throughout an entire day, as long as the subject stays below their aerobic threshold. We now address this more clearly in the limitations sections in line 414: “Likewise, activities beyond the aerobic threshold are not considered due to the challenges of obtaining ground truth energy expenditure values”.

Monitoring energy expenditure beyond the aerobic threshold is challenging because the buildup of lactic acid over time prevents accurate measurements with respirometry. Without accurate respirometry measurements to act as ground truth, wearable models cannot be trained or validated to estimate during anaerobic activities. We now address these ideas with an additional reference in the Methods, starting in line 598:

“These methods are capable of estimating energy expenditure for extended periods if the activity is within the aerobic threshold (15). Directly measuring instantaneous energy expenditure of the whole body is not possible. Respirometry cannot measure instantaneous energy expenditure because of significant delays in replenishing energy stored in the muscles, primarily due to mitochondrial dynamics, oxygen consumption, and blood circulation (50). Activities outside the aerobic threshold violate steady-state assumptions in these respirometry methods and thus cannot be monitored accurately.”

(50). P. Krstrup, A. M. Jones, D. P. Wilkerson, J. A. Calbet, J. Bangsbo. Muscular and pulmonary O₂ uptake kinetics during moderate and high intensity submaximal knee extensor exercise in humans. *The Journal of physiology* **587**, 1843-1856 (2009).

Reviewer #1 (Remarks to the Author):

Review Comments for "Sensing leg movement enhances wearable monitoring of energy expenditure"

This article suggested improved methods to objectively measure physical activities and estimates metabolic energy expenditure in real-time. The revised manuscript included additional data which can explain the reviewer comments. The data such as data processing, estimation methods of sensor are clearly explained and complemented the authors' logicity. Consequently, when the criteria of Nature Communications are taken into consideration, I would recommend "PUBLISH" of this work without additional revisions.

Recommendation : PUBLISH as is; no revisions needed.

Reviewer #2 (Remarks to the Author):

I thought the additions in the introduction to motivate how this wearable data-driven estimate was different from other data-driven approaches was helpful, but I think it needs to be further highlighted both in the hypothesis itself and in the discussion, otherwise it's hard to be convinced that this method is being tested against the start-of-the-art baseline. First, the hypothesis in line 107 doesn't mention anything about this distinction (It could read, "We hypothesized that using a data-driven method without subject-specific training data that relies on wearable measurements of lower-limb kinematics and segmenting input data by stride would estimate energy expenditure more accurately ..."). Second, the discussion could use a portion devoted to comparing this data-driven approach to other data driven methods that have been used such as [16].

One pretty minor thing is that the sentence in line 373 still sounds like it is a drawback of the data-driven method when in reality it is an advantage. It should read, "The Wearable System still had lower steady-state error than the Activity-Specific Model even given the fact that the Activity-Specific Model was based on ideal activity classification performed manually. This indicates that the Wearable System may extract additional information such as how the activity was performed, where changes in motion may relate to muscle-level energy expenditure."

Reviewer #3 (Remarks to the Author):

The authors addressed all my concerns

Reviewer #4 (Remarks to the Author):

My primary reservation with this manuscript during the first round of reviews was that the major conclusions of the work were not interesting to a broad audience. The authors response to that point is not particularly strong or convincing. In their revision, the authors did a lot of work to rebut or clarify the minute details of the data or methodology presented in the manuscript, but the unfortunate conclusion of the work is not very relevant or illuminating for device or systems engineers seeking to make next-generation, accurate wearables.

Reviewer #1 (Remarks to the Author):

Review Comments for “Sensing leg movement enhances wearable monitoring of energy expenditure”

This article suggested improved methods to objectively measure physical activities and estimates metabolic energy expenditure in real-time. The revised manuscript included additional data which can explain the reviewer comments. The data such as data processing, estimation methods of sensor are clearly explained and complemented the authors' logicity. Consequently, when the criteria of Nature Communications are taken into consideration, I would recommend “PUBLISH” of this work without additional revisions.

Recommendation : PUBLISH as is; no revisions needed.

We thank the reviewer for reviewing our manuscript and for their insightful comments that helped us improve this work.

Reviewer #2 (Remarks to the Author):

I thought the additions in the introduction to motivate how this wearable data-driven estimate was different from other data-driven approaches was helpful, but I think it needs to be further highlighted both in the hypothesis itself and in the discussion, otherwise it's hard to be convinced that this method is being tested against the start-of-the-art baseline. First, the hypothesis in line 107 doesn't mention anything about this distinction (It could read, "We hypothesized that using a data-driven method without subject-specific training data that relies on wearable measurements of lower-limb kinematics and segmenting input data by stride would estimate energy expenditure more accurately ..."). Second, the discussion could use a portion devoted to comparing this data-driven approach to other data driven methods that have been used such as [16].

We agree with the reviewer that it is important to differentiate that our method is accurate compared to the state-of-the-art methods and does not include subject-specific data. We have updated the manuscript with the suggested text.

Line 107: “We hypothesized that a data-driven method without subject-specific training data that relies only on wearable measurements of lower-limb kinematics segmented by stride, without subject-specific training data, could estimate energy expenditure more accurately than state-of-the-art methods during common activities including walking, running, stair climbing, and biking.”

We have added to the discussion to directly compare to previous data-driven approaches discussed in the introduction.

- 317: “Previous experiments found activity monitors to estimate energy expenditure with 30% error for wrist-worn devices (38) and 27% for hip-worn devices (39), suggesting

these locations do not capture motion related to energy expended by lower-limb muscles as accurately as sensors placed on the legs.”

- 305: “Previous smartwatch studies report similar errors from 35% to 93% when estimating energy expenditure for new subjects (30, 31, 39), supporting the idea that smartwatches have significant error.”
- 332: “The Wearable System was more accurate than wearable data-driven methods using a variety of sensors and subject-specific training data with errors of 14% to 27% (16, 28, 30), which would have approximately twice the error when evaluating new subjects (35).”

One pretty minor thing is that the sentence in line 373 still sounds like it is a drawback of the data-driven method when in reality it is an advantage. It should read, "The Wearable System still had lower steady-state error than the Activity-Specific Model even given the fact that the Activity-Specific Model was based on ideal activity classification performed manually. This indicates that the Wearable System may extract additional information such as how the activity was performed, where changes in motion may relate to muscle-level energy expenditure."

We agree with the reviewer that it should be clear that it is an advantage that the Wearable System was able to outperform a manually labeled Activity-Specific Model. We have updated the text with the reviewer’s suggestion.

380: "The Wearable System still had lower steady-state error than the Activity-Specific Model even given the fact that the Activity-Specific Model was based on ideal activity classification performed manually. This indicates that the Wearable System may extract additional information such as how the activity was performed, where changes in motion may relate to muscle-level energy expenditure."

Reviewer #3 (Remarks to the Author):

The authors addressed all my concerns

We thank the reviewer for their time in reviewing the manuscript.

Reviewer #4 (Remarks to the Author):

My primary reservation with this manuscript during the first round of reviews was that the major conclusions of the work were not interesting to a broad audience. The authors response to that point is not particularly strong or convincing. In their revision, the authors did a lot of work to rebut or clarify the minute details of the data or methodology presented in the manuscript, but the unfortunate conclusion of the work is not very relevant or illuminating for device or systems engineers seeking to make next-generation, accurate wearables.

We respectfully disagree. Our results have been of great interest to engineers seeking to make next-generation wearables. In addition, our study makes contributions that will be of interest to a broader audience, including members of the fields of biomechanics, machine learning, electromechanical engineering, and medicines, and others from diverse readership of *Nature Communications*. The main contribution is an inexpensive and accurate approach to estimate energy expenditure, which we expect to have broad appeal because it could be used to overcome challenges in accurately monitoring physical activity at large scale, studying relationships between activity and health, and managing weight. The system is open-source and can be replicated easily, facilitating application to products that address these challenges. This work also makes a scientific contribution through rigorous experiments to select the sensors that relate most closely to energy expenditure. We have made all of these contributions clear in the revised manuscript.